# Integrated regulatory models for inference of subtype-specific susceptibilities in glioblastoma

Yunpeng Liu[1,2,3], Ning Shi[4], Aviv Regev[1,2,3], Shan He[4] & Michael T Hemann[1,2,3,*]

## Abstract

Glioblastoma multiforme (GBM) is a highly malignant form of cancer that lacks effective treatment options or well-defined strategies for personalized cancer therapy. The disease has been stratified into distinct molecular subtypes; however, the underlying regulatory circuitry that gives rise to such heterogeneity and its implications for therapy remain unclear. We developed a modular computational pipeline, Integrative Modeling of Transcription Regulatory Interactions for Systematic Inference of Susceptibility in Cancer (inTRINSiC), to dissect subtype-specific regulatory programs and predict genetic dependencies in individual patient tumors. Using a multilayer network consisting of 518 transcription factors (TFs), 10,733 target genes, and a signaling layer of 3,132 proteins, we were able to accurately identify differential regulatory activity of TFs that shape subtype-specific expression landscapes. Our models also allowed inference of mechanisms for altered TF behavior in different GBM subtypes. Most importantly, we were able to use the multilayer models to perform an *in silico* perturbation analysis to infer differential genetic vulnerabilities across GBM subtypes and pinpoint the MYB family member MYBL2 as a drug target specific for the Proneural subtype.

**Keywords** cell state plasticity; gene essentiality inference; glioblastoma multiforme; subtype-specific gene regulation; transcription regulatory networks
**Subject Categories** Cancer; Computational Biology; Methods & Resources
**Mol Syst Biol. (2020) 16: e9506**

## Introduction

Glioblastoma multiforme (GBM) is the most common type of primary brain tumors, accounting for 80% of primary malignancies in the brain (Hanif *et al*, 2017). Despite being a rare disease, affecting less than 10 in 100,000 adults globally per year (Hanif *et al*, 2017), GBM remains one of the most clinically challenging types of tumor, with a dismal median survival of ∼ 15 months after diagnosis (Koshy & Mccarthy, 2014). Current treatment options include surgery, radiotherapy, and chemotherapy (temozolomide), which have been shown to extend patient survival but are not curative (Weller *et al*, 2005). Targeted therapies against growth factor receptors, PI3K/AKT/mTOR and MAPK pathways as well as cell cycle control are either currently under clinical trials or have shown little or no efficacy (Li *et al*, 2016; Zhao *et al*, 2017). Immunotherapies, including engineered chimeric antigen receptor T cells (CAR-T cells) against tumor antigens (IL-13Rα2, HER2, EGFRvIII etc.), have shown sporadic successes in animal models (Krenciute *et al*, 2017) and recent clinical trials (Rourke *et al*, 2017; Ahmed *et al*, 2020), yet factors including the highly immunosuppressive microenvironment in the brain and target antigen loss have posed significant obstacles to favorable responses.

Key therapeutic challenges in targeting GBM stem from high levels of heterogeneity within and across tumors, as well as the anatomically protected location of the tumors. In a pioneering effort to understand GBM heterogeneity, Verhaak *et al* (2010) performed multi-omics profiling of GBM tumors that uncovered four major molecular subtypes based on transcriptome signatures, named Classical, Neural, Proneural, and Mesenchymal. These four subtypes, which have recently been revised to three due to the Neural subtype likely arising from normal neuronal tissues in tumor margins (Wang *et al*, 2017), display distinct mutational landscapes as well as responses to radiotherapy. Interestingly, recent work on mapping the clonal organization of patient tumors (Sottoriva *et al*, 2013) as well as single-cell transcriptome profiling of GBM tumor samples have shown that expression signatures of the Classical, Proneural, and Mesenchymal subtypes are manifested at the single-cell level (Patel *et al*, 2014; Wang *et al*, 2017), suggesting that individual tumors fall into each of the three subtypes due to the fact that the majority of tumor cells within the tumor exhibit the corresponding subtype's signature. More recent single-cell studies have revealed additional layers of intra-tumoral heterogeneity due to the co-existence of immune cells (Darmanis *et al*, 2017) as well as plasticity of transcriptional programs within the same tumor (Neftel *et al*, 2019). These observations imply that a key route to dissecting GBM intra- and intertumoral heterogeneity is understanding the transcription regulatory networks that give rise to the different molecular subtypes and cell states. Despite the highly challenging nature of

1 Department of Biology, Massachusetts Institute of Technology, Cambridge, MA, USA
2 MIT Koch Institute for Integrative Cancer Research, Cambridge, MA, USA
3 Broad Institute of MIT and Harvard, Cambridge, MA, USA
4 School of Computer Science, University of Birmingham, Birmingham, UK
*Corresponding author. Tel: +1 6173241964; E-mail: hemann@mit.edu

such a task, a reasonable starting point is to infer the regulatory code of subtype-/cell state-specific expression programs. Pinpointing the transcription factors that are responsible for shaping distinct GBM transcriptional subtypes would provide a mechanistic view of the source of heterogeneity and, more importantly, allow systematic inference of susceptibilities in each subtype of GBM that would in turn facilitate the design of targeted therapy.

Several computational models have been proposed to understand global transcription regulation in mammalian cells so far. These models infer genome-wide regulatory links between TFs and target genes primarily using one or more of the following strategies: (i) expression association methods, (ii) analysis of physical binding of TF to promoters and enhancers, and (iii) regression models. Association between the expression of TFs and target genes are quantified by either co-expression metrics (Stuart *et al*, 2003; Langfelder & Horvath, 2007; Gaiteri *et al*, 2014; Liu *et al*, 2014; Aibar *et al*, 2017) or mutual information (Margolin *et al*, 2006; Lachmann *et al*, 2016) to infer potential regulatory relationships. However, these methods do not allow direct inference of causal relationships in transcription regulation, and association-based links usually need to be filtered based on additional evidence. Moreover, the parameters in these models do not contain direct information on both the magnitude and directionality of TF regulation of a given target, despite the fact that some of them can be interpreted as magnitude in a probabilistic fashion [e.g., ARACNe (Margolin *et al*, 2006; Lachmann *et al*, 2016)] or using rank-based scores [e.g., GENIE3 (Huynh-Thu *et al*, 2010)]. In TF physical binding-based models, ChIP-seq or chromatin accessibility datasets are used in combination with TF motif databases to determine regulatory links (Gerstein *et al*, 2012; Neph *et al*, 2012; Marbach *et al*, 2016). However, these methods also fail to explicitly model the magnitude of regulation. Linear regression models of transcription regulation assign coefficients that describe both the directionality in which and the relative extent to which each TF regulates each target gene (Setty *et al*, 2012; Li *et al*, 2014; Pearl *et al*, 2019). However, these models do not allow biologically meaningful interpretations of the inferred regulatory relationships, as gene expression regulation acts in a nonlinear fashion. Plaisier *et al* (2016) proposed an integrated genomic and transcriptomic model, SYGNAL, for dissecting GBM-related causal regulatory networks and predicting drug targets. The SYGNAL model utilizes a combination of somatic mutation profiles, inferred TF physical binding map and gene co-expression to infer TF and microRNA (miRNA) regulatory relationships, filters for experimentally supported edges, and uses network edge orientation to infer causality in the network. While the SYGNAL pipeline highlights key regulatory links (e.g., IRF1-IKZF1) in GBM and generates high-confidence inference on GBM-specific drug-miRNA combinations, it relies heavily on a thorough compendium of experimental evidence which may not be readily available for other diseases or biological processes. In addition, such a largely binary network model does not allow inference of quantitative changes in transcriptome and cellular phenotype in response to perturbations. Thus, a common issue insufficiently addressed by current methods for modeling transcription regulatory networks is the lack of parameterization that permits functional interpretation of the regulatory links and predictive modeling of gene expression and cellular phenotype. Such a feature is important for identification of

key regulators that are essential for the survival of cells in a given state and are thus potential drug targets.

To dissect the TF regulatory circuitry underlying the heterogeneity of GBM and construct a model that facilitates generation of actionable hypotheses for targeting different GBM subtypes, we integrated multiple lines of data and assembled a novel computational pipeline, Integrative Modeling of Transcription Regulatory Interactions for Systematic Inference of Susceptibility in Cancer (inTRINSiC). We combined two TF binding networks computationally inferred from tissue-specific chromatin landscape data and TF binding motifs, and parameterized edges using a nonlinear regression model built from thermodynamic description of transcription regulation. Fitting GBM gene expression data from the Cancer Genome Atlas (TCGA) to our model, we constructed subtype-specific transcription regulatory networks that explain gene expression variation and provide a mechanistic view of differential transcription factor activity. Importantly, we were able to predict each GBM subtype's dependency on each transcription factor by integrating our transcription regulatory model with protein signaling networks and gene essentiality data in the DepMap project (Meyers *et al*, 2017; Tsherniak *et al*, 2017) and show that perturbing the expression of a subset of transcription factors may provide subtype-specific therapeutic benefits.

## Results

### Nonlinear regression accurately models subtype-specific gene expression in glioblastoma multiforme tumor samples

To model transcription regulation in different subtypes of glioblastoma multiforme, we first constructed a candidate network of putative TF-gene regulatory interactions and then parameterized each interaction by fitting a nonlinear regression model to patient tumor gene expression profiles (Fig 1A and B). In order to infer a candidate set of brain tumor-specific regulatory pairs, we leveraged two sources of information: (i) DNaseI hypersensitivity profiles of brain tumor cell lines, which chart open chromatin regions that may be accessible to transcription factors and (ii) the JASPAR transcription factor binding motif database (Sandelin *et al*, 2004). We assigned each TF to each potential target gene by modeling the probability of a given TF binding to each open chromatin region using the Protein Interaction Quantification (PIQ) algorithm (Sherwood *et al*, 2014). We then expanded this set of candidate TF-target pairs by including edges from a previously constructed tissue-specific regulatory network for the central nervous system (Marbach *et al*, 2016)—the organ system giving rise to GBM development. Additional details on data processing and parameter selection for the network construction process are provided in Materials and Methods. The above procedures result in a list of candidate TF-target gene pairs that are binary and unsigned. Next, we inferred the strength and sign of regulation for each of these pairs using a nonlinear regression model that describes each target gene's expression as a sum of basal levels and additive regulatory effects (in logarithmic space) from its regulator TFs. Here, individual regulatory effects from TFs are based on a biophysical model of transcription regulation (Bintu *et al*, 2005), and we assume independent and saturable action of each TF. The parameter for each TF's regulatory effect on each target gene, $F$,

indicates the sign and strength of the regulatory interaction, where an $F$ value greater than 1 implies that the TF activates the target gene, an $F$ value between 0 and 1 implies repression of the target's expression by the TF, and an $F$ value of 1 denotes the absence of regulatory interactions between a given TF-target pair.

To infer subtype-specific $F$ values, we first classified TCGA GBM samples into different subtypes using the nearest centroid method where the subtype centroids are computed as in Verhaak *et al* (2010). To account for the recent discovery that the Neural subtype may be an artifact of normal tissue "contamination", we also used a new classification scheme by Wang *et al* (2017) comprised of only the Classical, Proneural and Mesenchymal subtypes, and compared the assignment of subtype labels. We find that the identity of over 80% of samples are conserved in each remaining subtype and that the Neural subtype samples were redistributed into the other three subtypes in the new classification scheme (Fig EV1B). We estimated $F$ values for each TF-gene pair in each GBM subtype by fitting the regression model to subtype-specific expression profiles in the TCGA GBM dataset and imposing an L2-like regularization term that penalizes large absolute values of parameters and prevents overfitting, and used 5-fold cross-validation to select the best hyperparameters for regularization strength (Fig EV1C). After running the regression model on the original and new classification schemes (omitting the Neural subtype samples in the original scheme), we compared the $F$ value profiles for each TF between the old and new labeling schemes and found that there is significant correlation between the two in all three subtypes (Fig EV1D). This suggests that the regression pipeline is robust with respect to the differences in sample numbers in each subtype. We will hereafter use the samples from all subtypes, except the spurious Neural subtype samples (likely representing a mixture of multiple cell states), from the original TCGA classification system for subsequent analyses.

Our nonlinear regression models are capable of accurately predicting GBM tumor gene expression with high subtype specificity, as shown by an inter-subtype expression prediction experiment where a subtype mismatch between the model and the data would result in a significant loss in accuracy of prediction (Fig 1C). Additionally, $F$ value profiles estimated from the bulk tumor data used in this study showed a significant correlation with those derived from subtype-specific single-cell expression profiles (Patel *et al*, 2014) (Fig 1D), indicating that neither the dataset nor our models were significantly biased by artifacts from bulk cell mixtures. We therefore performed all of our downstream analysis in this study using the TCGA bulk expression dataset, based on its high gene coverage and sample abundance compared with sparse single-cell data. Additionally, we compared models built from an independent RNA-seq set of 172 GBM samples from the Chinese Glioma Genome Atlas (CGGA) project (Wang *et al*, 2015; Liu *et al*, 2018) that were classified using the same method as we did for TCGA samples and found that there is a significant overlap between the regulatory edges inferred by the model on the TCGA and CGGA data. This overlap occurs despite vastly different sample sizes and expression profiling platforms (Figs 1E and EV1F), suggesting that the regression models are also robust with respect to the size and type of expression data used.

A key artifact that may diminish the power of this regression model is that its capability of capturing TF-target gene regulatory parameters may be limited to transcription factors showing high expression variability. We show that this is not the case by plotting the mean absolute $\log_2$-$F$ values for transcription factors against the coefficient of variation in their expression values (Fig EV1G), which revealed a non-monotonic relationship between mean regulatory strength and expression variability. Additionally, such a pattern is in agreement with the idea that TFs with extremely high regulatory potentials (high absolute $\log_2 F$ values, such as E2F family transcription factors E2F2 and E2F8) are normally maintained at stable expression levels, whereas those with medium regulatory strengths (such as TWIST1 and, interestingly, several HOX family genes) are variably expressed—possibly to mediate responses to stimuli and facilitate cell state changes. Figure 1F–H shows clustered heatmaps of $F$ value profiles of TFs with top average regulatory strengths as indicated by absolute $\log_2 F$ values. Several of these TFs, including GATA1 and SP1, have been suggested to be involved in the progression and invasiveness of glioblastoma cells (Guan *et al*, 2012). Importantly, the same gene is often co-regulated by a cluster of TFs, suggesting that our regulatory models may be able to capture interactions among TFs, which we will explore in the following sections of this paper.

## Gene expression modeling captures known interactions among transcription factors

We next asked whether the $F$ values obtained from our nonlinear regression models are consistent with known biology of transcription regulation. We hypothesize that TFs with correlated regulatory profiles, i.e., $F$ value vectors, are more likely to be interaction and/or co-regulatory partners (Fig 2A). To test this, we computed a robust correlation metric between each pair of regulatory profiles using Random Sample Consensus (RANSAC) outlier detection (Fischler & Bolles, 1981) and examined the top correlated pairs. The RANSAC algorithm is used to efficiently estimate the dominant correlation structure within each pair of $F$ value vectors, while guarding against extreme $F$ values that may substantially skew correlation calculated with canonical Pearson's correlation coefficient. Here, we first focused on TF pairs that are consistently correlated across GBM subtypes, which indicate that they may be instrumental for brain-specific transcription regulation or that they are part of the core transcription regulation machinery in the cell. Indeed, several well-characterized universal TF interactions emerge from the highly correlated pairs, an example of which is MYC and MAX, a pair of transcription co-activators for a large number of target genes across multiple tissue types. As shown in Fig 2B, MYC and MAX show tightly correlated regulatory profiles across all three GBM subtypes. Interestingly, on target genes where MYC shows repression (pink points), the correlation of its $F$ values with those of MAX tends to be diminished. This is in line with the known biology of MYC where instead of MAX it partners with the transcription factor MAD (not modeled in this analysis) to repress target gene expression (Amati, 1994; Grandori *et al*, 2000). When we set increasingly stringent cutoffs for correlation coefficients, we found that the top 1% correlated pairs show a significant enrichment of annotated interactions in the BIOGRID database (Chatr-aryamontri *et al*, 2015; Fig 2C, Fisher's exact test $P$-value $= 1.655 \times 10^{-12}$). Consistently, the observed level of enrichment significantly exceeds the range of distribution estimated from random sampling of TF-TF

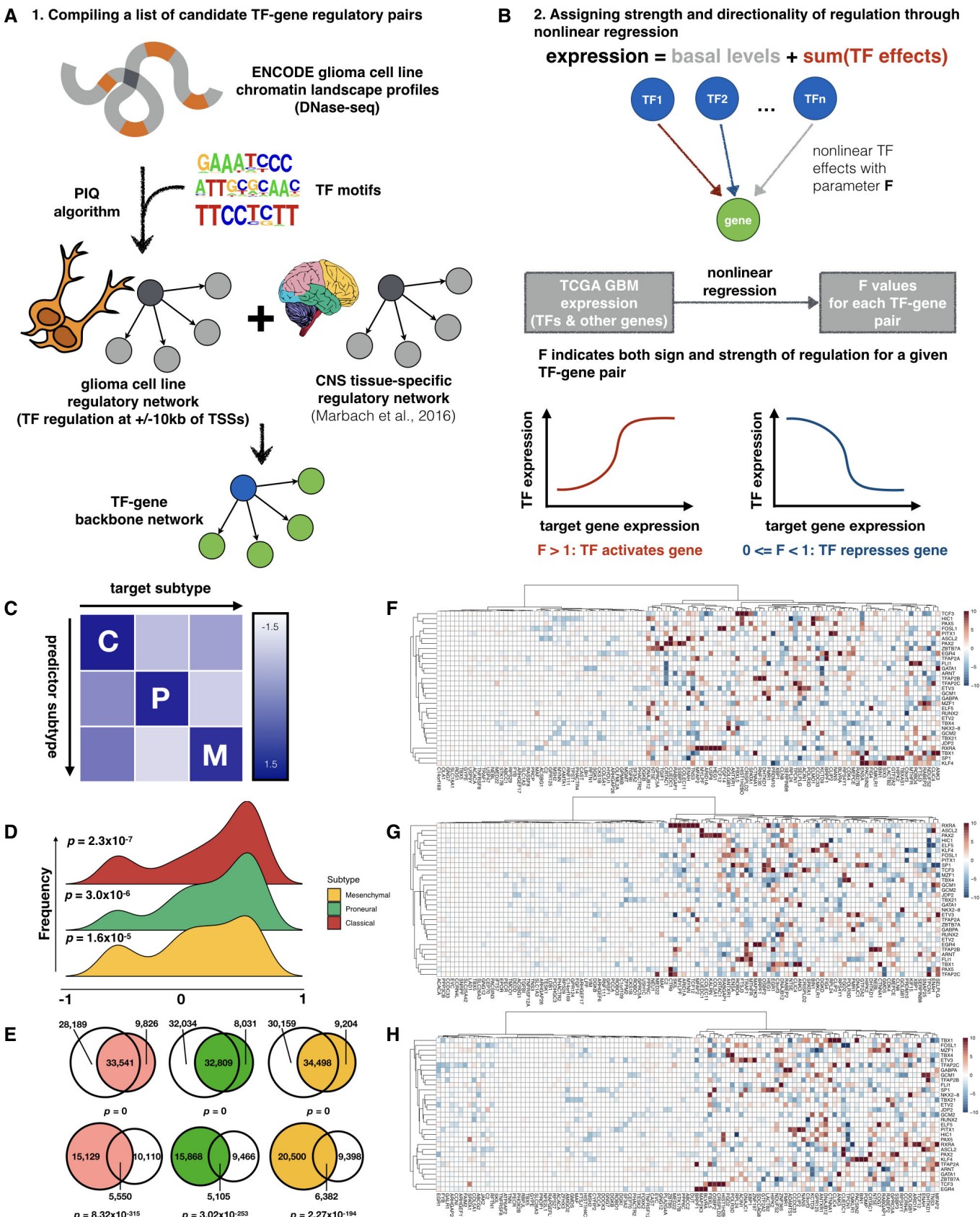

**Figure 1.**

**Figure 1. Modeling subtype-specific transcription regulation in glioblastoma tumor samples.**

A    Schematic of workflow for compiling a candidate network for GBM-specific transcription regulation.

B    Nonlinear regression assigns magnitude and directionality of transcription regulation for each transcription factor-target gene pair through a parameter, *F* value.

C    Inter-subtype prediction matrix, where the regression model from each subtype along a column is used to predict gene expression in another subtype along a row, and scaled average of the median symmetric mean absolute percentage error (sMAPE) values across top variable genes is visualized as heatmap colors.

D    Distribution of correlation coefficients between *F* value profiles obtained from bulk (TCGA) and single-cell datasets.

E    Venn diagrams showing overlap of edges between regression models built from TCGA and CGGA where a transcription factor shows up-regulation (upper row) or down-regulation (lower row) of its target genes. Shown from left to right are Classical (red), Proneural (green), and Mesenchymal (orange) subtypes, respectively, and corresponding CGGA subtypes are colored in gray. Fisher's exact test *P*-values for Venn diagrams are labeled correspondingly.

F–H  Heatmaps of representative subsets of *F* values for the 3 GBM subtypes (Classical, Proneural, and Mesenchymal, respectively). Rows and columns correspond to transcription factors and target genes with top absolute $\log_2 F$ values, respectively.

pairs (1,000 iterations, simulated *P*-value = 0, Fig 2D). Figure 2E shows a network of known BIOGRID interactions among the top 1% correlated TF pairs, where a few well-characterized pairs/hubs, including CTCF-YY1, HIF1α-Sp1, and the histone acetyltransferase EP300 that regulate a broad range of target genes, are highlighted in blue.

As a comparison, we have also constructed GBM subtype-specific regulatory networks using ARACNe (Margolin *et al*, 2006; Lachmann *et al*, 2016), a mutual information-based method for inferring regulatory links based on gene expression. We found that there is minimal correlation between the regulatory parameters inferred by ARACNe (i.e., estimated mutual information) and *F* values from our pipeline (Fig EV2B). This suggests that ARACNe and inTRINSiC may be exploring vastly different types of regulatory interactions. To test if this is the case, we mined for TF-TF interactions by computing robust correlation coefficients among the parameters estimated by ARACNe for each pair of TFs. Indeed, inTRINSiC infers a substantially larger set of TF interactions that significantly overlap with those inferred by ARACNe (Fig EV2C). When we examined the number of known BIOGRID interactions captured by the two methods, as well as those captured by GENIE3, another commonly used network inference tool that ranks TF-target regulatory edges using random forest regression importance metrics (Huynh-Thu *et al*, 2010) [utilized as part of the single-cell regulatory network construction pipeline SCENIC (Aibar *et al*, 2017)], we find that interactions inferred from all three methods show significant enrichment of known interactions at increasingly stringent cutoffs for correlation between regulatory profiles (*F* value or mutual information vectors, Fig EV2D). Thus, inTRINSiC shows comparable detection power for regulatory interactions as two other state-of-the-art tools for building regulatory models. In addition, inTRINSiC potentially expands the inferred TF-TF interaction space compared with ARACNe in a manner that warrants further experimental interrogation. A summarized comparison between inTRINSiC, ARACNe, and GENIE3 is provided in Table EV1.

**Transcription factor repurposing may be responsible for shaping subtype-specific transcriptomes**

After confirming that our model accurately explains gene expression variation in GBM and that the regulatory parameter *F* delineates key features of transcription regulation, we proceeded to investigate whether GBM subtypes show distinct regulatory landscapes, and if so, to what extent altered transcriptional regulation explains differential expression profiles across subtypes.

To mine for subtype-specific regulatory profiles, we performed a permutation experiment where subtype labels were randomly shuffled and new subtype-specific regression models were fitted for each iteration. We looked for TF-target pairs that showed subtype-specific regulatory parameters that are unlikely to be observed in permuted data (see Materials and Methods for details). Such an analysis revealed many TFs that displayed distinct regulatory parameters in one GBM subtype compared with the other two for the same set of target genes. We hereafter term such altered behavior "repurposing" of transcription factors and deemed TFs with the majority of its targets showing *F* values unique to a single subtype the "signature TF" of that subtype. An example of TF repurposing is shown in Fig 3A, where MXI1 is a Proneural subtype signature TF as defined above. It can be seen in the heatmap of $\log_2 F$ values that MXI1 represses a subset of targets only in the Proneural tumor samples. Plots of the expression of one of MXI1's targets, *ITGA5*, against that of *MXI1* itself are shown in Fig 3A. Consistent with the trends in *F* values, there is a negative correlation between the expression of *MXI1* and *ITGA5* only in the Proneural subtype. To define a regulatory signature for each of the three GBM subtypes, each TF that showed significant single-cell expression levels was assigned a signature subtype. Since it is likely that the same TF may contribute to the signature regulatory profiles of multiple subtypes, we computed a score quantifying the levels of participation of each subtype's signature TFs in that subtype, shown as heatmaps in Fig 3B. Indeed, despite high exclusivity of a subset of signature TFs (DDIT3 and ELK4 in the Mesenchymal subtype, for example), many signature TFs show comparable participation in the signature of at least one other subtype (STAT5B, MYC, and STAT3, etc.). This is consistent with the fact that the latter type of TFs regulate a broad range of cellular functions in a tissue-nonspecific way.

Transcription factor activity could potentially shape the differential gene expression landscapes observed across GBM subtypes in one of three ways: (i) through altered TF expression *per se*, (ii) through differential regulatory strength and directionality (manifested as altered *F* values), or (iii) a combination of these two mechanisms. Since we observed *F* value profiles that are unique to each subtype, we were particularly interested in the second mechanism. To examine the extent to which GBM subtype-specific transcriptomes are explained by subtype-specific TF regulatory parameters, we identified genes within each subtype's expression signature whose TFs showed concomitant repurposing in that subtype (see Materials and Methods for details). Interestingly, we found that expression signature genes are significantly enriched for genes that underwent differential regulation by TFs in all but one subtype (Fig 3C). Additionally, when we computed the extent to which

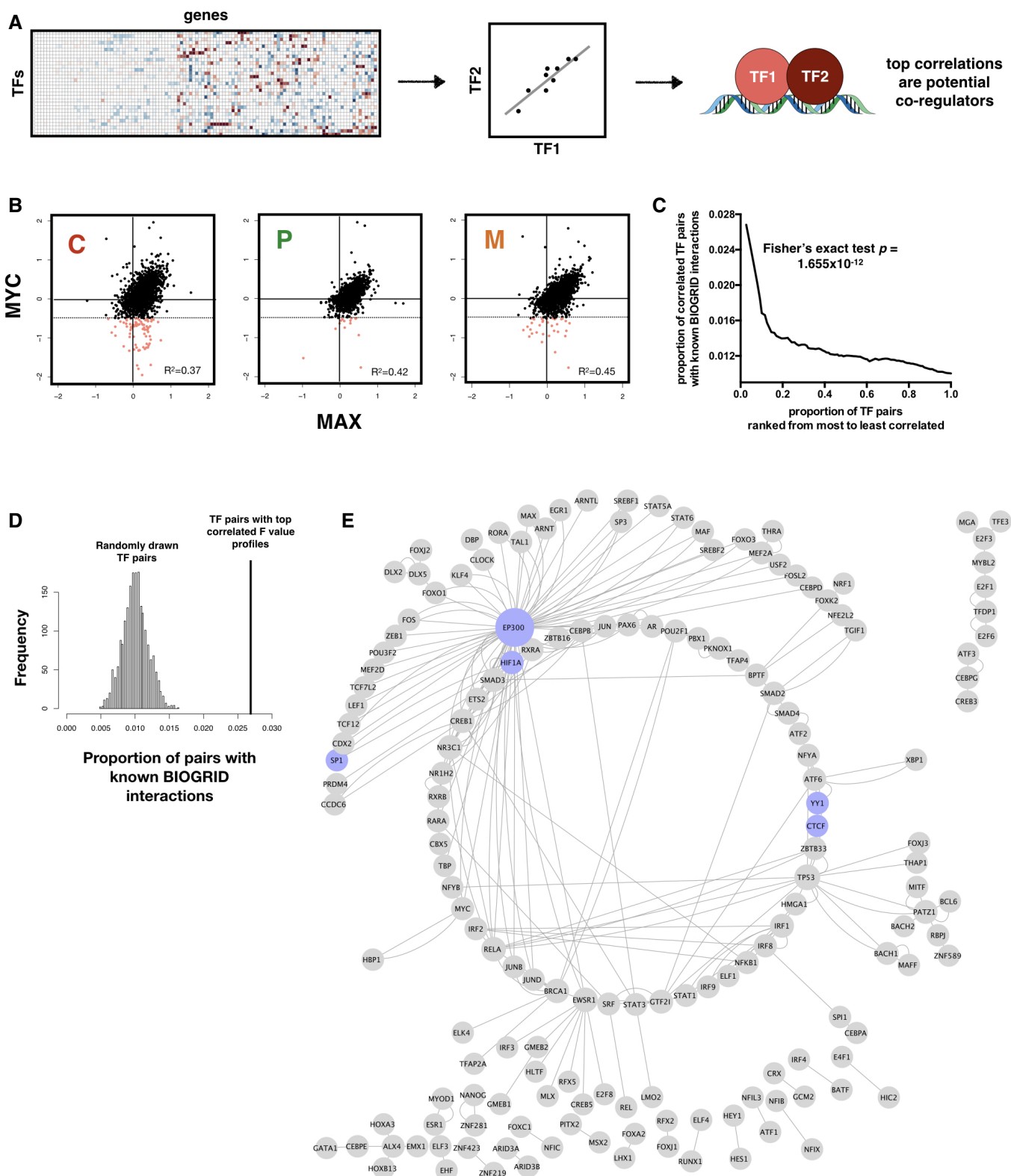

**Figure 2.**

target gene expression is correlated with that of their regulators, we found that there is a significant decrease in the mean correlation coefficient when comparing genes which are differentially regulated

by TFs across subtypes (manifested as subtype-specific $F$ values) with those which are not (Fig 3D), further supporting the idea that TF repurposing serves as a significant source of differential gene

**Figure 2.   Nonlinear regression models capture known interactions among transcription factors.**

A   Schematic of workflow for inferring TF-TF interactions.

B   Scatter plot showing consistent correlation between the regulatory profiles (*F* values) of MYC and MAX across all three subtypes. Pearson correlation coefficient ($R^2$) values are shown. Regions where MYC shows negative regulation of gene expression (hence likely to be interacting with MAD rather than MAX) and MAX shows no significant regulation are shaded pink, with threshold for negative regulation shown as dotted horizontal lines.

C   TF-TF pairs with top *F* value correlations are enriched for known interactions in the BIOGRID database. *P*-value for Fisher's exact test of proportion of known interaction pairs among top 1% correlated TFs compared to selecting all possible interaction pairs is shown.

D   Distribution of prediction precision of known BIOGRID interactions from 2,000 randomly drawn TF pairs (histogram) compared with those predicted from TF pairs with top 1% *F* value correlation (vertical line). Simulated *P*-value is 0.

E   Network representation of captured known BIOGRID interactions from top 1% *F* value correlation coefficients. A subset of nodes participating in well-characterized interactions, including CTCF-YY1 and dense interaction subnetwork of EP300, is colored in blue.

expression. Note that the Classical subtype is an exception here, likely due to a significantly lower number of genes involved in the analysis or the possibility that TFs may change regulatory behavior across different expression levels due to negative feedback mechanisms (Chalard *et al*, 2009).

An important feature of our transcription regulatory models is that they allow prediction of gene expression changes when the expression levels of TFs are perturbed. To further examine the roles which signature TFs play in shaping and maintaining subtype-specific transcription landscapes, we next asked whether perturbation of certain TFs can shift the transcriptome profiles of one subtype of GBM to another. Such cell state changes may be particularly relevant in GBM, as GBM cells have been shown to display phenotypic plasticity where cells transition from the Proneural to a Mesenchymal phenotype (Fedele *et al*, 2019), and that patient GBM cells from the same genetic origin could give rise to progeny harboring divergent transcriptional states (Neftel *et al*, 2019). To see if our models can recapitulate transcriptional plasticity in GBM, we designed a recursive algorithm to simulate the propagation of changes in regulatory effects throughout the transcription regulatory network upon knocking down a given TF (see Materials and Methods for further details and pseudocode for the algorithm). Having applied such a perturbation algorithm to the transcription factor STAT3 (a Mesenchymal signature TF, see heatmap in Fig 3B), we compared the transcriptome profiles of patient samples before and after perturbation in a 2-D embedding (Fig 3E). STAT3 has been suggested to act as a master regulator of the Mesenchymal phenotype in GBM (Carro *et al*, 2010). We hypothesized that loss of STAT3 expression may induce a shift away from the Mesenchymal phenotype. Indeed, as shown by orange arrows in Fig 3E, many Mesenchymal samples shifted toward the Classical cluster upon perturbation of STAT3 (red arrows), implying that a subset of Mesenchymal GBM samples lose their established cell states. Interestingly, a few Proneural samples were redirected (green arrows) toward the Mesenchymal cluster. This may be due to the observation that STAT3 showed remarkable signature participation in both the Mesenchymal and Proneural subtypes (Fig 3B heatmap). Other examples of transcriptomic shifts toward another subtype in TCGA patients in response to perturbation of signature TFs include HOXA1 and NEUROG1 (Fig EV2E and Table EV2). Interestingly, when we inspect the subtype specificity of top shift-inducing TFs, we observe the largest transcriptome shifts in Mesenchymal subtype samples (Fig 3E, right panel), suggesting that the Mesenchymal subtype may represent a metastable state that is hyper-sensitive to diverse perturbations. Taken together, our signature and perturbation

analyses further support the idea that signature transcription factors maintain cell states and subtype identity in GBM.

**Subtype-specific regulatory profiles in glioblastoma multiforme may be explained by altered functional partnering among transcription factors**

How do transcription factors switch regulatory behavior on the same set of target genes across different cell states? A signature analysis of correlation among regulatory profiles similar to that of *F* values offers mechanistic insight into such alterations. Here, we transition from regulatory signatures, which describe how each TF differentially regulates each *target* gene in each subtype, to co-regulatory signatures, which capture the correlation structure among the regulatory capacities of *pairs of TFs* at multiple target genes in each subtype. Specifically, we computed RANSAC correlation coefficients among *F* value vectors of transcription factors in each subtype and looked for correlated TF pairs which are unique to each subtype. An example of subtype-specific co-regulatory profiles that emerge from the analysis is shown in Fig 4A, where the *F* values of the transcription factors MYB and MSX2 are only correlated in the Mesenchymal subtype.

We defined a set of co-regulatory signature TFs for each subtype in a similar way to that used to extract regulatory signatures and again calculated a participation score for each signature TF across the three subtypes (Fig 4B). Here, the number of signature co-regulatory TFs in each subtype does not seem to correlate with that of signature regulatory TFs as shown in Fig 3B, suggesting that additional mechanisms apart from changes in co-regulatory partners are involved in TF repurposing.

For a global view of differential TF-TF partnering, we plotted circos (Krzywinski *et al*) diagrams representing top correlations between the *F* value profile of each co-regulatory signature TF with that of other TFs (Fig 4C, upper panel). It seems that the overall architecture of such a co-regulatory network is largely conserved across GBM subtypes (Fig 4C, upper panel and Fig EV2A). When we focused on individual signature TFs, however, local rewiring of the co-regulatory network emerged, for example in the case of the Proneural co-regulatory signature TF, MXI1 (Fig 4C, lower panel). Interestingly, MXI1 is also a regulatory signature TF as determined in the previous section. In fact, there is a significant overlap between the regulatory and co-regulatory signature TF sets (hypergeometric test *P* = 0.011, with overlapping TFs shown in Venn diagram in Fig 4D), suggesting that the altered behavior of several TFs could be due to differential coordination with other TFs. To see if this was the case, we examined the correlation of *F* values across

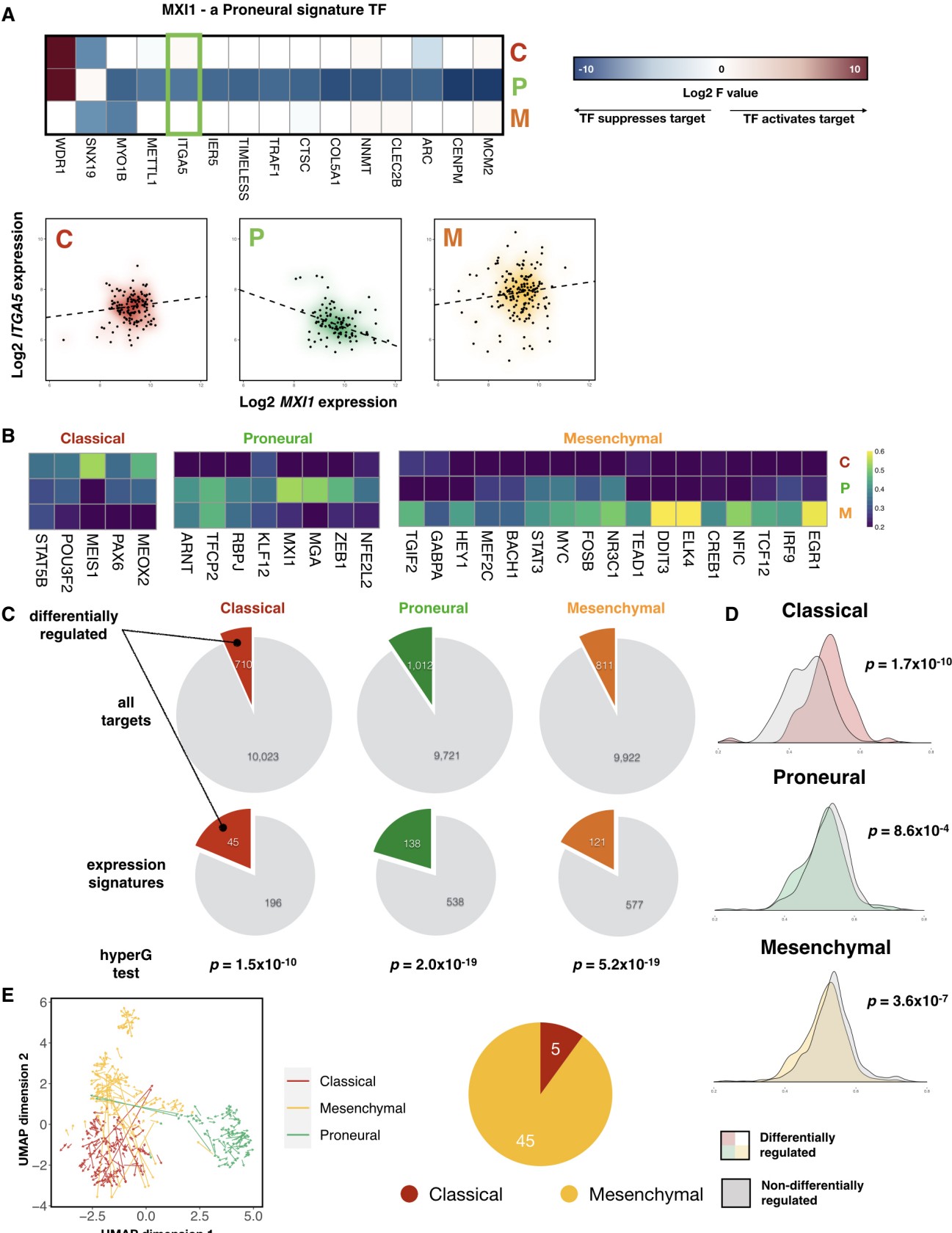

**Figure 3.**

**Figure 3.  GBM tumor samples exhibit unique transcription factor regulatory profiles that shape subtype-specific gene expression landscapes.**

A  Left panel: heatmap of signature *F* values on a $\log_2$ scale for MXI1, a Proneural subtype signature transcription factor. Right panel: gene expression plots of *ITGA5*, a signature target gene of *MXI1* (highlighted in green box in left panel), against that of *MXI1*, across the three GBM subtypes. Density clouds are overlaid onto scatter plots, and a dashed linear regression line is shown for each subtype.

B  Heatmaps showing participation of each subtype's signature TFs in the corresponding subtype signature. Color scale corresponds to the proportion of TF-target gene pairs that show consistent signature behavior with the corresponding subtype signature among all TF-target pairs that show any signature behavior.

C  Pie charts showing proportion of target genes that show differential regulation by TFs, comparing all target genes (upper panel) with those that are differentially expressed among the subtypes (i.e., subtype expression signatures, lower panel). *P*-values from hypergeometric tests for enrichment of differentially regulated target genes among expression signatures in each subtype are shown.

D  Kernel density estimations of average correlation coefficients between target genes and their corresponding transcription factors. *P*-values for Mann–Whitney tests comparing each pair of distributions are shown.

E  Left panel: perturbation of STAT3 induces Mesenchymal samples to shift toward the Classical subtype cluster. Each arrow points from the TCGA sample before perturbation to the same sample after perturbation, projected onto a 2-dimensional Uniform Manifold Approximation and Projection (UMAP) space. Right panel: pie-chart showing distribution of the subtype showing the largest overall shift among the 50 TFs that induce the largest subtype-specific shift.

target genes which undergo differential regulation across subtypes. Indeed, we found that several co-regulatory TFs showed significantly different distributions of correlation coefficients with other TFs at target genes which were differentially regulated (Fig EV6). Figure 4E shows the example of MXI1 (left panel), where additional correlations, or partnerships, with other TFs are gained in the Proneural subtype. This suggests that analyses of *F* values could pinpoint changes in the coordinated action of TFs at specific target genes that, in turn, altered the regulatory strength and/or directionality of particular TFs. For TFs that did not show subtype-specific co-regulatory partnerships at differentially regulated target genes (for example the glucocorticoid receptor NR3C1, a Mesenchymal signature TF, see Fig 4E right panel), possible mechanisms may include trans-repression, where a TF can achieve indirect repression of target genes by associating with other activator TFs and prevent them from binding to their respective regulatory regions, as in the case of NR3C1(Ray & Prefontaine, 1994).

**Integrated transcription regulation-protein signaling network models enable *in silico* screening for gene essentiality and infer new therapeutic targets in glioblastoma multiforme**

Having established that our transcription regulatory models could both accurately capture gene expression variation and provide mechanistic insights into the mode of action of transcription factors, we next sought to build a novel, integrated pipeline for predicting phenotypic output of TF perturbations beyond transcription and dissecting subtype-specific dependencies in glioblastoma. A schematic of this subroutine and its relationship with other components of the inTRINSiC pipeline is shown in Figs 5A and EV5. Specifically, the effects of transcription regulation (i.e., gene expression values) are overlaid onto a protein signaling network, where a random walk-based algorithm termed Exponential Ranking (Traag *et al*, 2010) is employed to estimate the activity level of each protein (see Materials and Methods for a detailed description of the algorithm). Our method differs from existing protein activity scoring algorithms such as VIPER (Alvarez *et al*, 2016) in that it does not rely on strengths of transcription regulation as a "readout" for protein activity, i.e., instead of assessing how much a regulator ultimately changes the expression of other genes (as in VIPER, which is coupled to output from ARACNe), we use gene expression values (computed by the nonlinear regression models of inTRINSiC) and direct protein–protein signaling interactions to estimate the relative activity levels of proteins based on the strength and nature of the

interactions. When comparing protein signaling activity estimated from VIPER and that from our exponential ranking method, we found that exponential ranking was able to estimate protein activity changes for all 3,132 proteins covered by our signed, weighted model whereas VIPER only covered 218 proteins deemed as master regulators by the algorithm (Fig EV2F). Additionally, among the covered interactions, VIPER did not capture some of the well-established signaling activities, such as the EGFR/STAT3 and EGFR/KRAS axes, where EGFR is expected to activate these two downstream targets (Fig EV2G and H).

Using such a framework for simulating perturbation of TF expression as well as information flow in the cellular circuitry, we fitted our regulatory models to 32 brain tumor cell lines (using a training-validation set split of 24-8 and 4-fold cross-validation) in the CCLE cell line collection (Barretina *et al*, 2012), which can each be assigned an expression subtype based on proximity to centroids extracted from TCGA tumor data (Fig EV3A and B, see Materials and Methods for details). We first performed Exponential Ranking on these 32 cell lines to estimate protein activity levels. To infer effects of knocking down individual TFs, we then perturbed the expression of each TF using our recursive algorithm. A new set of protein scores were then computed based on new expression levels obtained from the perturbation. Next, we trained an elastic net regression model for each perturbed TF to find a sparse linear combination of proteins, the changes in whose activity best predict the TF's gene essentiality, as quantified by depletion scores in the DepMap genetic screen dataset (Materials and Methods).

Our training-validation pipeline identified a set of protein activity changes, following *in silico* perturbation of TFs, that could potentially predict TF essentiality in individual cell lines. We tested these predictions on eight independent CCLE cell lines that were not included in our initial training-validation set. Our regression models were able to predict essentiality scores that were highly consistent with experimentally determined scores for the majority of TFs examined in this study (Fig 5B showing TFs ranked by the inverse root mean square error of predicted essentiality scores, and Fig 5C, with mean correlation deviating significantly from 0, Wilcoxon rank sum test *P*-value = $4.99 \times 10^{-29}$). Figure 5C also shows the four TFs with lowest prediction error (Fig 5B) and highest correlation between predicted and DepMap essentiality scores. Additionally, essentiality scores predicted by our pipeline show consistent subtype specificity with experimentally determined DepMap essentiality scores (Fig 5D). In other words, TFs that we identified previously as core contributors to subtype specificity also show subtype

essentiality. Interestingly, patterns in the regression coefficients (Fig EV4A and B) imply that a subset of TFs share common predictor proteins, suggesting that the effects of perturbing these TFs are likely to be mediated by common pathways. On the other hand, the same TF's knockdown effects may be mediated by a multitude of proteins, which is consistent with the notion that perturbing a single

TF may result in pleiotropic effects throughout the intracellular signaling network that collectively reduce fitness of the cell.

Having confirmed that the effect of knocking down at least a subset of the TFs can be accurately predicted by combinations of changes in protein scores, we next asked if such a model could predict the effects of TF knockdown in patient tumor data, where it

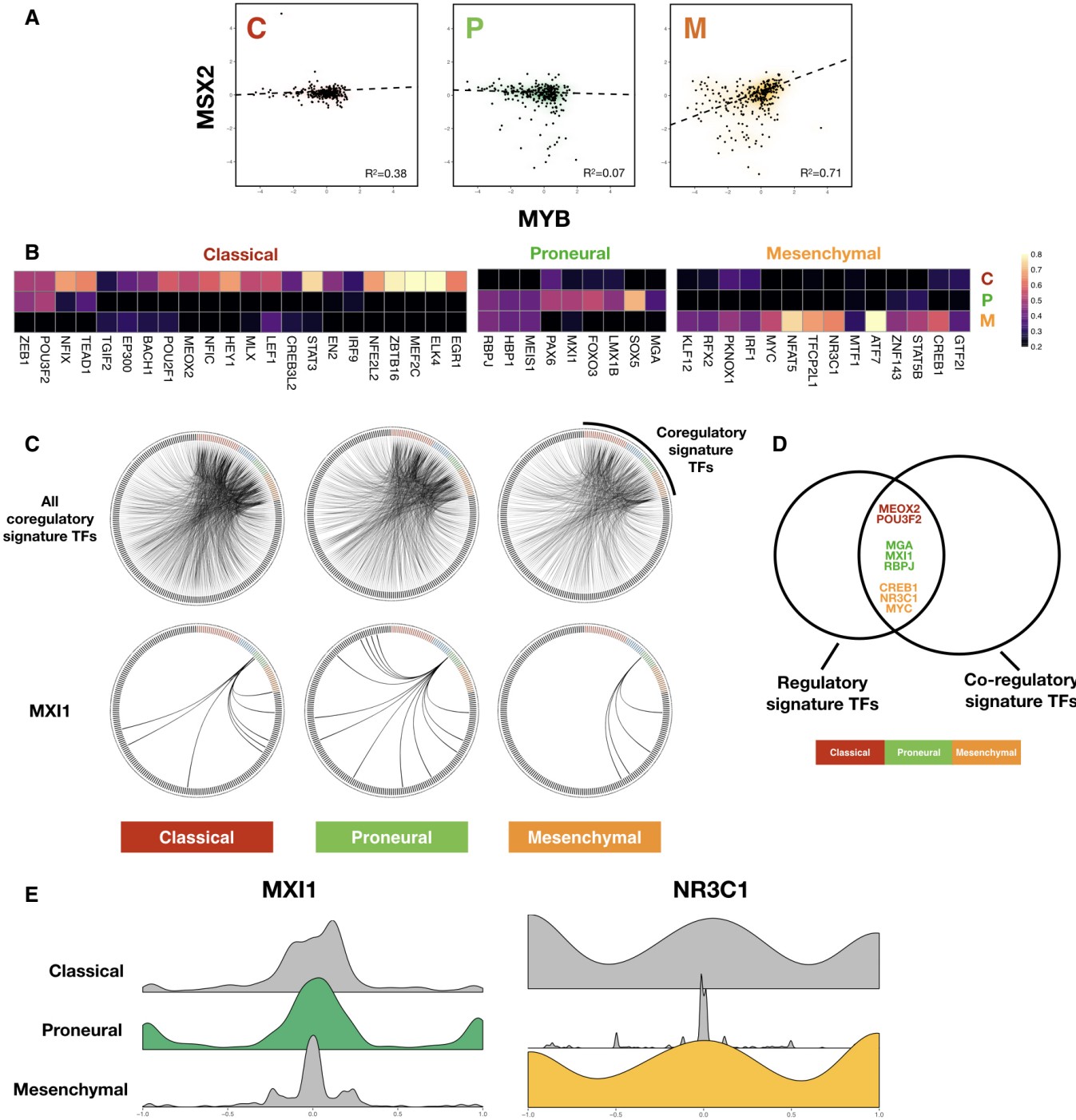

**Figure 4.**

**Figure 4. Subtype-specific co-regulatory interactions among TFs may explain differential regulatory behavior.**

A   Scatter plot showing consistent correlation between *F* value profiles of MSX2 and MYB across each subtype. Robust regression lines (dashed lines in each panel) as well as Pearson correlation coefficient ($R^2$) values are shown. Density clouds are overlaid onto scatter plots to highlight trends.

B   Heatmaps showing participation of each subtype's signature TFs in the corresponding subtype TF-TF co-regulatory signature. Note that here instead of *F* values of TF-target regulation, signatures are derived from correlation coefficients of *F* value profiles between pairs of TFs across subtypes. Color scale corresponds to the proportion of TF-TF co-regulatory pairs that show signature behavior consistent with the corresponding subtype signature among all TF-TF pairs that show any signature co-regulatory behavior.

C   Upper panels: circos diagrams showing all top co-regulatory partners (gray nodes along the outer circle) inferred for subtype signature TFs (colored nodes along the outer circle). TF pairs with *F* value correlation coefficients larger than 0.8 are visualized as links. Lower panels: same TFs but only showing links originating from MXI1, a Proneural subtype signature TF.

D   Venn diagram showing overlap between regulatory and co-regulatory signature TFs, colored according to GBM subtypes.

E   Kernel density plots of correlation coefficients of signature TFs MXI1 (Proneural) and NR3C1 (Mesenchymal) with other TFs at differentially regulated target genes.

is intractable to perform individual unbiased genetic screens. We normalized the CCLE and TCGA expression datasets to remove batch effects and ran the perturbation—regression pipeline described above on the CCLE subset to obtain regression coefficients for each protein. Using the same coefficients, we computed the expected knockout/knockdown depletion score (i.e., gene essentiality score) for each TF in individual TCGA GBM tumor samples. We observed that the knockdown of certain TFs confers significantly different predicted levels of survival disadvantages across subtypes, examples of which include *MYBL2* (Fig 5D) and *NFE2* (Fig EV4C, see Table EV3 for a full list of predicted subtype-specific survival disadvantages for each TF). We found six transcription factors with predicted subtype-specific essentiality scores that also show consistent subtype-specific negative correlation between their expression levels and tumor patient survival (Fig 5E). Here, we focus on *MYBL2*, whose knockdown is significantly more detrimental to the Proneural subtype than to the other subtypes *in silico* (one-way analyses of means *P*-value $1.51 \times 10^{-61}$, *df* = 2.0, *df*2 = 215.3). Consistently, we only see a negative correlation between *MYBL2* expression and survival in the Proneural subtype samples (Fig 5F). In addition, log-rank tests of survival of *MYBL2*-high versus *MYBL2*-low patients in the TCGA dataset (as demarcated by median expression within each subtype) show that *MYBL2*-low patients show significantly higher survival rates only in the Proneural subtype (log-rank test *P* = 0.035, Figs 5G and EV4D). We observed a similarly significant trend in an independent set of 172 GBM samples from the Chinese Glioma Genome Atlas (CGGA) project (Wang *et al*, 2015; Liu *et al*, 2018), where low MYBL2 levels correspond to higher survival only in the Proneural subtype (log-rank test *P* = 0.028, Figs 5G and EV4D). Interestingly, our *F* value correlation analysis showed that MYBL2 displays high correlation with the Proneural signature TFs TFCP2 and MXI1, suggesting that subtype-specific TF interactions may be key to maintaining cell state and viability in the corresponding subtype. The above analyses demonstrate that the inTRINSiC pipeline is capable of identifying potential subtype-specific drug targets for GBM, for example MYBL2 in the Proneural subtype.

## Discussion

We propose a computational framework, inTRINSiC, for integrating epigenomic, transcriptomic, protein interaction, and genetic perturbation data to dissect tumor heterogeneity in glioblastoma multiforme and use this framework to systematically infer subtype-

specific vulnerabilities. inTRINSiC serves as a powerful, tractable platform for distilling large-scale omics data into candidate genes that are critical for different cell states and generating actionable hypotheses for targeting tumors in a subtype- and even patient-specific way.

We first showed that transcription regulation by TFs can be quantitatively described using biophysical models, which despite simplifying assumptions was still able to accurately capture gene expression variation across GBM subtypes (Figs 1C and EV1C) as well as TF-TF co-regulatory interactions (Fig 2C and D), which are not explicitly factored into the model. In addition, bulk tumor expression data did not seem to significantly confound our analysis despite cell state mixture effects, as models built from single-cell expression data showed significant consistency with those from bulk data (Fig 1D). Note that a small subset of transcription factors (e.g., FOSB and RXRA in Classical and Proneural subtypes) showed near-zero or negative correlation between bulk and single-cell *F* value profiles, possibly due to immune cell and/or normal brain tissue infiltration of tumors in bulk samples. However, these inconsistencies did not seem to significantly confound our downstream analyses, since the TFs that showed such behavior did not participate in the signatures discovered for each subtype or score as top essential TFs. We were also are able to uncover subtype-specific regulators, consistent with single-cell expression-based discoveries, for example DDIT3 in the Mesenchymal subtype (as seen in Neftel *et al*, 2019 and Fig 3B). Compared with existing regulatory network inference methods such as the information theory-based ARACNe package (Margolin *et al*, 2006; Lachmann *et al*, 2016), and the correlation-based models tailored for single-cell expression data by SCENIC (Aibar *et al*, 2017; see Table EV1 for a summarized comparison of methods and Fig EV2D for a comparison of performances), a distinct feature of the nonlinear regression method used in the inTRINSiC pipeline is that it explicitly models the quantitative regulation of steady-state gene expression by transcription factors. While such a model may be oversimplifying the intricate process through which gene expression is modulated and relies substantially on the availability of physical binding/chromatin accessibility and TF motif data, it yields easily interpretable parameters that directly correspond to the maximum fold change (as well as the directionality of change) a given TF can induce in its target gene expression. These parameters and their resulting quantitative output are not explicitly modeled in other pipelines. Additionally, even without direct modeling of higher-order interactions between TFs, the inTRINSiC pipeline is still able to capture TF-TF co-regulatory interactions that are not as effectively recovered in linear methods (Fig EV1C).

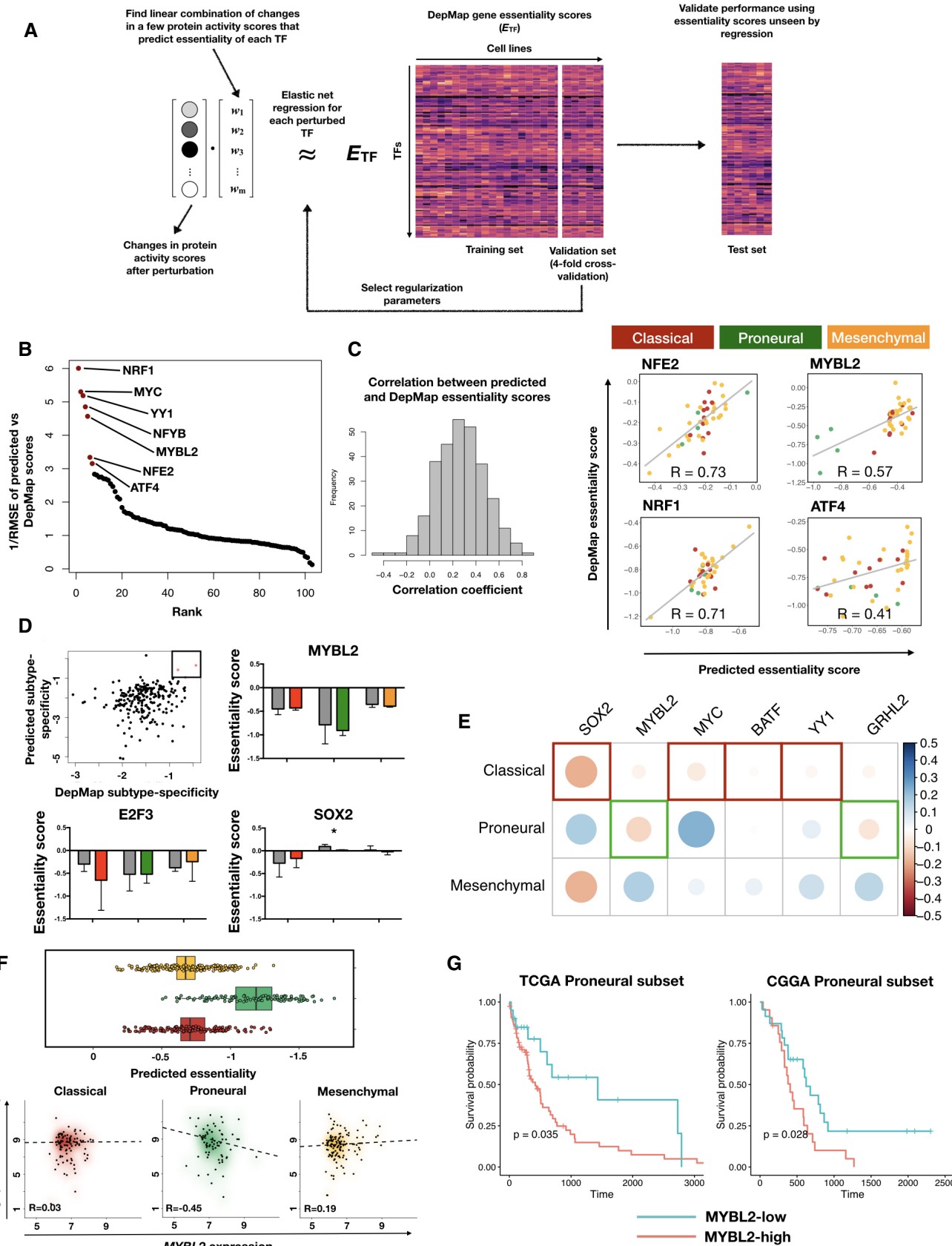

Figure 5.

**Figure 5. Integrated multilayer regulatory network model enables *in silico* perturbation and inference of subtype-specific drug targets in GBM.**

A  Schematic of *in silico* perturbation through multilayer network information flow simulation. Left: changes in gene expression induced by perturbation of a TF can be estimated through transcriptional regulatory models, and propagated to the protein signaling network, where a modified random walk algorithm scores signaling activity. Middle: iteratively perturb each TF and generate a matrix of perturbed signaling scores. Right: optimal "readout" of effects of TF perturbation on fitness can be learned through known genetic screening data such as DepMap gene essentiality scores.

B  Transcription factors ranked by the prediction accuracy in test set cell lines (*y* axis, plotted are 1 over root mean squared error values) of their essentiality when compared to DepMap scores.

C  Left panel: histogram of correlation between predicted and DepMap TF essentiality scores. Right panel: experimentally determined (DepMap) versus predicted gene essentiality scores for TFs where a linear combination of protein signaling activity scores could predict TF essentiality with high accuracy. Data points are color-coded according to subtype labels (red—Classical, green—Proneural and orange—Mesenchymal).

D  Upper left panel: subtype specificity of predicted versus DepMap essentiality scores. Three TFs with top consistent specificity values are highlighted in a box region. Remaining panels: bar plots showing mean ± standard error of the mean (SEM) of DepMap (gray) and predicted (color-coded according to subtypes as in (C)) essentiality scores, grouped by subtypes. # of cell lines in each subtype: Classical—11, Proneural—4, Mesenchymal—25. Paired Student's *t*-test *P*-values for each paired set of DepMap-predicted results are non-significant after Bonferroni correction except for the Proneural subtype in SOX2 perturbations (*$P$ = 0.044).

E  Heatmap of correlations between TF expression and TCGA patient survival in each subtype. Shown are the six TFs that are consistent in terms of the most dependent subtype predicted by inTRINSiC and the subtype showing the most negative correlation between TF expression and patient survival.

F  Box and jitter plots of predicted essentiality scores of MYBL2 in TCGA tumor samples grouped by subtype. Box plots show 25th, 50th, and 75th percentiles, and whiskers extend up to 90th and down to 10th percentiles. # of samples: Mesenchymal—165, Proneural—103, Classical—139.

G  Kaplan–Meier survival curves of TCGA and CGGA Proneural GBM samples, grouped by MYBL2 score (high versus low demarcated by median expression). # of patients in TCGA Proneural subset: MYBL2-high—47, MYBL2-low—48. # of patients in CGGA Proneural subset: MYBL2-high—24, MYBL2-low——25. Log-rank test *P*-values are shown. Survival curves for the remaining subtypes are provided in Fig EV4D.

It is important to note here that regulation by transcription factors is only one of several mechanisms of gene expression control, and other factors including miRNAs and long non-coding RNAs may also play significant roles. However, we did not include non-coding mRNAs in our current version of inTRINSiC pipeline for two reasons: (i) the effects of miRNA regulation are estimated to explain only 7–13% of overall gene expression variation (Vejnar & Zdobnov, 2012) and (ii) since miRNAs act primarily via repressing mRNA levels (Guo *et al*, 2010), regulation by miRNAs may be absorbed into TF regulation in our model where a subset of TF-target regulatory parameters may be due to indirect regulation of the target by TFs through expression of miRNAs (Guo *et al*, 2010). This does not preclude that certain miRNAs may represent critical nodes of regulation or that the inclusion of miRNA data in the future may refine our analysis. In our current version of inTRINSiC, we also did not explicitly model the effects of genetic alterations on transcription regulation. This was due to a lack of systematic catalogues documenting how specific mutations affect transcription factor functions. In addition, the relatively low frequency of mutations and their uneven distribution across subtypes would significantly diminish the statistical power in our regression models. However, we do expect the effects of mutations to be implicitly modeled by *F* values, as transcription factors whose regulatory activities are affected by mutations are likely to display altered *F* value profiles. In fact, our regulatory/co-regulatory signature analyses hint at effects potentially mediated by genetic alterations. For example, MYC, which shows a Mesenchymal-specific co-regulatory profile (Fig 4B), has been previously shown to be upregulated by EGFR signaling and induce transition to a mesenchymal phenotype (Dong *et al*, 2018). Interestingly, such a signature is not seen in other subtypes where the EGFR mutations are enriched (Verhaak *et al*, 2010).

An important feature of the regulatory models built from the inTRINSiC pipeline is that aside from differentiating between different GBM subtypes in terms of the behavior of each TF (Fig 3B), they also provide clues as to the potential mechanisms of the same TF switching regulatory behavior on the same set of target genes (Fig 4). Specifically, we found that despite conservation of the

majority of TF-TF partnering relationships (Fig EV2A), TFs that are part of a subtype signature tend to display local, differential partnering with other TFs across proximal regulatory regions of target genes, implying that the regulatory heterogeneity observed across GBM subtypes may be a result of differential physical clustering of TFs at promoters/enhancers. Importantly, it has been shown that the transcription factor STAT3 (a Proneural/Mesenchymal signature TF inferred by inTRINSiC) and its associated signaling network play key roles in promoting an immunosuppressive tumor microenvironment that may hinder immunotherapy (See *et al*, 2012; Jackson *et al*, 2011). It would thus be interesting to further investigate the multitude of co-regulatory relationships predicted by inTRINSiC. We demonstrated that the combinatorial changes in the action of TFs could explain a significant proportion of differential expression observed across GBM subtypes (Fig 3C and D).

Finally, we performed an *in silico* screen for each subtype's dependency on different transcription factors by simulating information flow from the transcription regulation layer to the protein signaling layer and learning combinatorial protein activity that best predicts known gene essentiality scores (Fig 5), and uncovered TFs that display GBM subtype-specific essentiality. Such a functionality is unique to the inTRINSiC pipeline due to predictive models of transcription regulation and is not encompassed by other integrative pipelines such as SYGNAL (Plaisier *et al*, 2016; Table EV1). Another feature of this second part of the inTRINSiC pipeline is that the method for protein signaling activity inference is uncoupled from that of transcription activity—we trimmed all edges that only belongs to the "transcriptional regulation" category in the signaling networks for this step such that TF regulatory effects are required to be modeled by the TF-target network construction part of inTRINSiC, and only signaling/post-translational regulation activities are explicitly modeled in the downstream protein network. Such a strategy ensures consistency with known signaling relationships and achieves a larger coverage of signaling proteins than methods that infer protein activity based on transcriptional "regulons", such as VIPER (Fig EV2F–H). The TFs deemed essential by inTRINSiC can in turn be prioritized for further experimental validation in relevant cell lines or patient derived xenografts (PDXs). Key advantages

of such an *in silico* perturbation system include obviation of *in vivo* screens in PDX samples which may not engraft well, and prediction of tumor susceptibility on an individual sample basis. Indeed, we describe a work flow by which data from functional studies on cancer cell lines, like the DepMap, can be readily applied to emerging patient tumor data from diverse sources to identify subtype-specific vulnerabilities. While these vulnerabilities are TFs, a set of proteins that are notoriously difficult to target using small molecules, this approach may also converge upon druggable downstream targets of TFs. Additionally, there has been significant promise in the development of new strategies to target transcriptional regulators and TF interactions (Bushweller, 2019).

In this study, we identified MYBL2 as a transcription factor essential for the Proneural subtype. Interestingly, in a separate analysis of GBM subtype-specific regulatory programs, MYBL2 was also identified as a signature regulator of the Proneural subtype (Setty *et al*, 2012). MYBL2 is a member of the MYB family of transcription factors that plays important roles in cell cycle progression and maintenance of cells in an undifferentiated state (Musa *et al*, 2017). Its functions oppose that of ASCL1—a transcription factor promoting neuronal differentiation—and putatively crucial to maintaining the Proneural transcription landscape (Narayanan *et al*, 2019), which is consistent with the observation that overexpression of ASCL1 *in vitro* (Narayanan *et al*, 2019) and knockdown of *MYBL2 in silico* (this paper) are detrimental to Proneural subtype cells. Additionally, the fact that the subtype-specific essential TF MYBL2 is also implied to be functional interaction partners with the corresponding subtype's signature TFs further supports the idea that our signature analyses are capable of capturing GBM subtype-specific biology.

The inTRINSiC pipeline extends beyond GBM and serves as a new paradigm for understanding disease heterogeneity and different cell states through an integrated, multilayer network of transcription and protein activity regulation (Fig EV5). It is a flexible platform in that additional layers of regulation, including chromatin modifications and enhancer regulations, can be readily factored into the transcription regulation models wherever tissue-specific data are available. Additionally, with improvements in the scale and coverage of newer techniques such as Perturb-seq and single-cell epigenomics, the inTRINSiC pipeline can be extended to distinguish the source of phenotypic heterogeneity at the single-cell level through modeling multiple layers of regulation in the same cell (Fig EV6).

# Materials and Methods

## Reagents and Tools table

| Reagent/Resource | Reference or Source |
|---|---|
| **Software** | |
| CRAN R version 3.6.3 or higher | https://cran.r-project.org/ (R Core Team, 2013) |
| RStudio version 1.2.5042 or higher | http://www.rstudio.com/ (RStudio Team, 2020) |
| Python 2.7 or higher | http://www.python.org |
| scikit-learn package version 0.22.1 or higher | https://scikit-learn.org/stable/index.html (Pedregosa *et al*, 2011) |
| MATLAB r2016a or higher with MATLAB Compiler (MCC) | https://www.mathworks.com/products/matlab.html |
| circos version 0.69 or higher | https://circos.ca (Krzywinski *et al*, 2009) |
| Cytoscape Version 3.7.1 or higher | http://www.cytoscape.org (Shannon *et al*, 2003) |
| GraphPad Prism version 6 or higher | https://www.graphpad.com/scientific-software/prism/ |
| Protein Interaction Quantification (PIQ) algorithm software package | https://bitbucket.org/thashim/piq/src/master/ (Sherwood *et al*, 2014) |
| **Resources** | |
| TCGA GBM gene expression & survival datasets | https://www.cancer.gov/tcga |
| CGGA GBM gene expression & survival datasets | http://www.cgga.org.cn/ (preprint: Zhao *et al*, 2020) |
| ENCODE glioma cell line DNaseI-seq datasets | https://www.encodeproject.org/<br>ENCODE Project Consortium (Dunham *et al*, 2012), ENCODE data portal (Davis *et al*, 2018)<br>Accession codes: ENCFF338CJE, ENCFF422GJX, NCFF899YRC, ENCFF175YAU, ENCFF251KOI, ENCFF397OZE, ENCFF001DWK, ENCFF001CAA, ENCFF001DWM |
| FANTOM5 CNS-specific regulatory networks | https://fantom.gsc.riken.jp/5/ (Lizio *et al*, 2019, 2015) |
| JASPAR motif database, fifth expansion | http://jaspar.genereg.net/ (Mathelier *et al*, 2016) |
| CCLE expression datasets (v2018q2) | https://portals.broadinstitute.org/ccle/data (Barretina *et al*, 2012) |
| DepMap gene essentiality dataset (v2018q2) | https://portals.broadinstitute.org/ccle/data (Tsherniak *et al*, 2017; Meyers *et al*, 2017) |
| STRING human protein interactions network v11 | https://string-db.org/ (Szklarczyk *et al*, 2019) |

**Reagents and Tools table** (continued)

| Reagent/Resource | Reference or Source |
|---|---|
| BIOGRID human protein interactions network v3.4.158 | https://downloads.thebiogrid.org/BioGRID/Release-Archive/BIOGRID-3.4.158/ (Oughtred *et al*, 2019) |
| FunCoup human protein interaction database v4.1 | http://funcoup.sbc.su.se/downloads/ (Ogris *et al*, 2018) |

## Methods and Protocols

A combined overview of the steps involved in the inTRINSiC modeling framework is shown as a schematic in Fig EV5. Here, we provide a step-by-step walkthrough of the procedures involved, from gene expression, epigenomics, motif and previously compiled tissue-specific network datasets to ultimately predicting transcription factor essentiality in different GBM subtypes.

### Reconstruction of the GBM transcription regulatory network

To obtain a backbone regulatory model that is specific to the tissue of origin in glioblastoma multiforme (GBM) and encompasses as many potential regulatory edges as possible, we assembled a transcription regulatory network model for brain tissue using publicly available tissue-specific models from the FANTOM5 project (Marbach *et al*, 2016) and augmented the network with another one inferred from motif and chromatin landscape information specific to brain tumor cells. For the FANTOM5 brain network, we used the union of all networks built from brain tissues covered in the project, retaining top 10% edges in each network. The cutoff was arbitrarily selected to control model size and did not significantly reduce coverage of expressed genes and transcription factors (see network statistics reported below). We describe details of the procedures for inferring the latter network in the following.

We first retrieved DNase hypersensitivity sequencing (DHS-seq) datasets in four human brain tumor cell lines (Daoy, H4, A172, and M059J) from the ENCODE project (Davis *et al*, 2018). Analysis of expression data for these cell lines from the Cancer Cell Line Encyclopedia (CCLE) (Barretina *et al*, 2012) database shows that their transcriptomes fall close to those of TCGA patient samples and resemble those of one or more GBM subtypes (Fig EV3A and B), and are thus phenotypically relevant to our model. Next, combining DHS-seq data that chart open chromatin landscapes in these cell lines and position weight matrices (PWMs) from 579 motifs in the JASPAR motif database (Sandelin *et al*, 2004), we inferred potential regulatory edges from 518 transcription factors (TFs) to target genes using the Protein Interaction Quantitation (PIQ) algorithm (Sherwood *et al*, 2014). Briefly, we applied the algorithm to each cell line's DHS-seq data to assign likelihood scores of proteins (TFs) binding to open chromatin regions, and selected interactions within the top 10% of all scores to obtain a motif-region map for that cell line. The percentage cutoff was chosen to roughly match the size of the brain-specific network and minimize biases introduced by gene coverage (Fig EV1A). Motif-region maps are then converted to TF-gene maps using the closest gene method, where a TF is assigned to the gene(s) closest to its open motif(s) within a ± 10,000 bp window from the transcription start site. Note that this method is likely to favor selection of promoters and proximal enhancers. The above pipeline resulted in four separate regulatory networks based on motif binding likelihood, and the union of three networks was

computed for the final brain tumor-specific DHS network. Here, we excluded the network built from M059J data due to the significantly smaller number of edges, probably due to insufficient sequencing coverage.

Prior to all analyses, we filtered the networks for genes whose expression data are available. GBM microarray gene expression data were retrieved from the Cancer Gene Atlas (TCGA) data portal as of July 2014. The expression dataset has not been significantly revised since then and we chose microarray expression due to a larger subset of patients covered. The RMA-normalized expression matrix consists of 544 patient samples and 12,042 genes. After filtering, the unionized DHS network from four brain tumor cell lines consists of 494,860 edges from 424 TFs to 10,388 genes. When we compared the DHS network with that of the brain-specific FANTOM5 network constructed by Marbach *et al* (2016) (containing 201,095 edges from 460 TFs to 10,050 genes after expression filtering), we see that there is only a small overlap between the two (~ 5% of the DHS network covered by FANTOM5 network). This does not seem to be due to systematic biases in the coverage of regulatory edges by the DHS data, as the DHS-based network showed an enrichment of FANTOM5 edges with increasing binding score cutoffs for controlling network size (Fig EV1A). The final backbone network we used for regression is the union of these two filtered networks, containing 653,800 edges from 518 TFs to 10,733 genes.

### GBM subtype-specific parameterization of regulatory interactions using nonlinear regression

To understand regulatory heterogeneity in GBM, we set out to assign biologically meaningful parameters to the backbone regulatory network and select edges that display strong regulatory capacities using a nonlinear regression model. Our model is built upon a thermodynamic description of transcription regulation by Bintu *et al* (2005) and assumes the following conditions: (i) each transcription factor (TF) acts upon each target gene in a nonlinear, saturable association with its own expression, (ii) effects of transcription regulation of different TFs on the same target gene are independent of each other and (iii) effects of different transcription factors are multiplicative (additive in logarithmic space). Denote the expression vector (comprised of $K$ samples) of a given target gene $i$ and each TF $j$ ($j \in \{n_{i1}, n_{i2}, \ldots, n_{ir}\}$) of its $r$ candidate regulatory TFs (as dictated by the backbone network) as $\overrightarrow{y_i}$ and $\overrightarrow{x_i}$, and let $x_{i0}$ be the basal level of gene expression without regulation from any of the candidate TFs, we have:

$$\log_2 \overrightarrow{y_i} = \log_2 \overrightarrow{x_{io}} + \sum_{j \in \{n_{i1}, n_{i2}, \ldots, n_{ir}\}} \log_2 \frac{1 + F_{ji} \overrightarrow{x_j}}{1 + \overrightarrow{x_j}} \tag{1}$$

where vector operations are element-wise. The single parameter for each TF-gene pair $F_{ji}$ can be thought of as the regulatory capacity of TF $j$ for gene $i$ and theoretically determines the maximum

amount of fold change TF $j$ can induce upon the expression of gene $i$. Specifically: (i) if $F_{ji} > 1$, gene $i$ is activated by TF $j$, (ii) if $F_{ji} < 1$ gene $i$ is repressed by TF $j$, and (iii) if $F_{ji} = 1$, TF $j$ has no regulatory effect on gene $i$.

The above model is fitted to expression data from all samples in each GBM subtype using the limited memory-BFGS method (Liu & Nocedal, 1989) in MATLAB with an L2-like penalty term, where subtype class labels are assigned to each of the TCGA GBM samples using the nearest centroid method based on expression cluster centroids published in the original works of Verhaak *et al* (2010). To filter for truly subtype-specific TF-target relationships, we shuffled the subtype labels 30 times and re-computed the F values using the same number of samples per label group, and looked at the unshuffled *F* values showing deviation from the shuffled F values of at least 5 standard deviations. To classify all central nervous system (CNS) tumor cell lines in the CCLE DepMap dataset into relevant GBM subtypes, we first normalized CCLE cell line expression data against the TCGA GBM expression dataset using the RemoveBatchEffects function in the R limma package and standardized all expression data to have a mean of zero and standard deviation of 1. We observe that the CCLE samples mix well with TCGA samples after normalization (Fig EV3A and B). We then used a nearest centroid method to classify CCLE cell lines using the subtype signature genes and their reference expression centroids as determined by Verhaak *et al* Similarly, for inferring F values from the CGGA dataset, we first performed batch effect removal using the R limma package to normalize $\log_2$-transformed CGGA RNA-seq expression data to have similar distribution to that of the TCGA microarray expression data before assigning subtype labels and inputting subtype-specific expression to the regression pipeline. We lay out details of hyperparameter selection for the nonlinear regression in the following step.

### Hyperparameter selection for parameterized GBM subtype-specific regulatory network

To prevent overfitting the nonlinear regression, we imposed an L2-like penalty term that results in the following objective function:

$$\arg \min E(\vec{F}_i) = \sum_{k=1,2,\dots,K} \sum_{j \in \{n_{i1}, n_{i2}, \dots, n_{ir}\}} (\log_2 y_{ik} - \log_2 y_{ik,obs})^2 + \lambda \sum_{j \in \{n_{i1}, n_{i2}, \dots, n_{ir}\}} (\log_2 F_{ji})^2,$$

where the expression estimate of gene $i$ in sample $k$, $y_{ik}$, is computed using equation 1 in the Materials and Methods section of this paper, and logarithm values are used for numerical stability. The hyperparameter lambda determines the strength of the penalty on large $F$ values that may dominate the regulatory profile of a given transcription factor (TF). We chose a final lambda value of 0.1 which achieved a low gene expression value prediction error in 5-fold cross-validation as well as the highest area under curve of a precision-recall curve constructed by computing the amount of known TF-TF interactions captured by each model (Fig EV1C, see the next section for details of TF-TF interaction prediction). As a comparison, we computed the same metrics for a simple linear

model (trained using L2-regularized regression) for predicting gene expression, and Fig EV1C (black curve) shows that the nonlinear model is superior to the linear model across a broad range of regularization strengths.

### Extraction of correlated regulatory profiles

As previously mentioned, our regression model for parameterizing transcription regulation does not explicitly model interactions, i.e., effect of each TF on the target gene is independent of one another. Nonetheless, we anticipated correlation structures to emerge from the $F$ value profiles of TFs since we fit a regression model to the expression of each gene independently. To compute a robust correlation strength for each pair of $F$ value vectors (each corresponding to the regulatory profile of a TF across all target genes), we first eliminated target genes that were not regulated by either one of the TFs to reduce zero inflation. We then eliminated potential "outliers" by using the Random Sample Consensus (RANSAC) regression algorithm implemented in the scikit-learn Python package (Pedregosa *et al*, 2011), using a minimum of 80% of samples to determine outliers. As discussed in the main text, the RANSAC method helps capture the core correlation structure and prevents large biases originating from a small subset of extreme $F$ values. Correlation was then computed as the Pearson correlation coefficient between the trimmed $F$ value vectors if their lengths are greater than 3, and set to 0 otherwise.

### Inferring protein activity from an integrated transcription regulation-signaling network

We first constructed a protein signaling network containing edges between protein pairs with annotated activating or inhibitory interactions in two databases: STRING (Szklarczyk *et al*, 2015) and FunCoup (Schmitt *et al*, 2014). The former database provides signed, unweighted links, and the latter provides likelihood scores for interaction between each protein pair, inferred through Bayesian integration of multiple lines of evidence. The final signaling network consists of 20,473 signed and directed links among 3,132 proteins in the STRING network that have been assigned a likelihood score by the FunCoup network. To infer protein signaling activity, we used a node ranking method, termed Exponential Ranking (Traag *et al*, 2010), similar to the PageRank algorithm (Page *et al*, 1998). Specifically, the algorithm assigns a rank score to each node (representing a protein) in a network (in our case a protein signaling network) with both positively (activating) and negatively (inhibitory) weighted links by modeling the flow of "trust" across the network using discrete choice theory. The algorithm works by iteratively updating the "trust" scores (in this case protein activity scores) using the following equation:

$$p(t+1) = \frac{\exp \frac{1}{\mu} A^T p(t)}{|| \exp \frac{1}{\mu} A^T p(t) ||_1}$$

where $p$ is the vector of protein rank scores that will eventually converge to estimated protein activity scores, $A$ is the transition matrix, and $\mu$ is the sole parameter in the algorithm. In our implementation of the algorithm, we used the difference between the maximum and minimum of the transition matrix, $\max_{ij} A_{ij} - \min_{ij} A_{ij}$, for parameter $\mu$, which is within the recommended range that

guarantees convergence. To initialize the iterative procedure outlined above, we scaled expression data predicted by the regression models to have a sum of 1 as the initial $p(0)$ and augmented the transition matrix using expression values of pairs of interactors, i.e., $A = xx^{T}B$ where $x$ is a column vector of expression values and $B$ is the weighted adjacency matrix of likelihood scores described above. The $p$ vector is updated using the above equation until changes between two consecutive updates (in the form of Frobenius norm) are within a small tolerance ($1 \times 10^{-6}$). In all downstream applications, we transformed the final protein activity scores by taking the logarithm of all positive scores or that of the absolute values for negative scores.

For a fair comparison with VIPER, we used the above signaling network as the interactome, together with subtype-specific expression matrices, as input for the VIPER algorithm. We ran both exponential ranking and VIPER on unperturbed expression profiles as well as expression profiles where the expression of a (non-transcriptional) regulator in known signaling pathways (e.g., EGFR) was decreased, and compared the estimated protein activity of downstream signaling targets to inspect the performance of the two models on capturing post-translational activity.

### Modeling effects of gene expression perturbation in silico

To compute the effects of gene expression perturbation, in particular that of transcription factors, we developed a recursive algorithm to account for the hierarchical and feedback properties of transcription regulation, i.e., the downstream targets of a perturbed TF may regulate another layer of TFs and/or regulate said TF itself. Since our thermodynamics-based model of transcription regulation essentiality models steady-state expression levels, we use a recursive framework where we traverse and update the network of transcription regulation with a modified breadth-first search strategy that detects feedback loops and ensures that each TF's expression is updated only once. A key advantage of such an algorithm is that it takes into consideration that a target gene may also be a TF, and that a particular target gene can be regulated by multiple regulator TFs that may also be perturbed due to a cascade of upstream perturbations. Our algorithm is designed such that each TF can at most be perturbed once, i.e., any given TF will not be perturbed again if it has already been perturbed and there exists a path between that TF and its regulator (i.e., a feedback loop). This ensures maximal propagation of TF regulatory effects without creating infinite loops. The pseudocode for such an algorithm, implemented in the "hPerturb" function, is as follows:

```
function hPerturb(TFToPerturb) {
Find all targets of TFToPerturb;
if (targets of TFToPerturb contain other TFs) {

Flag the TFs within the target set that should not be
perturbed using the subroutine doNotPerturb()
outlined below;

Flag all target genes that are not in the above list and
recalculate their expression with updated TF
expression values using the nonlinear regression
model;
```

```
If there are no target TFs that are perturbed in the
previous step, return; otherwise flag these TFs as
already perturbed and recursively perturb the
downstream targets of these TFs by calling the hPerturb
() function on each of them.
} else {

Recalculate all target gene expression with updated TF
expression values using the nonlinear regression model
and return.
}
```

Subroutine for determining TFs that should not be perturbed:
```
function doNotPerturb(topmostPerturbedTF,
TFToPerturb) {

Find the target TFs which have already been perturbed
and show feedback regulation of TFToPerturb;

Find the target TFs whose regulators that reside on the
path from the topmost perturbed TF
(topmostPerturbedTF, i.e. the argument in the very
first call to the hPerturb() function) have not all
been perturbed yet;

Return the union of the above two sets.
}
```

Using models built from brain tumor cell line expression data in the CCLE collection (Barretina et al, 2012), we predicted new target gene expression profiles after simulating knockdown for each of the 518 TFs considered in our models and used the new gene expression estimates for computing updated protein activity scores as described in the previous section. We then built an elastic net regression model to predict essentiality scores for each of the perturbed TFs based on changes in protein activity upon perturbation. Specifically, we model the TF-gene essentiality scores as determined by genetic screens in the DepMap project (Tsherniak et al, 2017; Meyers et al, 2017) as a linear combination of changes in protein activity scores and fitted the model using L1 and L2 regularization terms to decrease overfitting and ensure sparsity. Hyperparameters in the elastic net regression, i.e., strengths of L1 and L2 regularization, were determined for each TF using 4-fold cross-validation with 80% of available CCLE lines (for a total of 32).

To ensure that CCLE and TCGA expression profiles are comparable and that models trained using CCLE datasets can be applied to predicting gene essentiality in individual TCGA patient tumors, we performed batch effect correction using the RemoveBatchEffects functionality in the limma package in R (Ritchie et al, 2015). All perturbations are performed on normalized, batch effect-corrected data.

## Data availability

All code associated with the inTRINSiC pipeline and analyses in this paper, as well as instructions on how key pipeline procedures should be called, can be accessed from the following GitHub repository: https://github.com/yunpengl9071/inTRINSiC.

**Expanded View** for this article is available online.

## Acknowledgements

We thank Azucena Ramos, Catherine Koch, and James Dongjoo Ham for useful discussions and suggestions for the paper. This project was funded in part by the NCI (R01 CA233477-01). Y. Liu acknowledges funding support from the MIT Koch Institute Graduate Fellowship. S. He would like to thank the Southern University of Science and Technology (SUSTech) for the sabbatical visiting program. Y. Liu and A. Regev acknowledge the Howard Hughes Medical Institute for funding support. Y. Liu and M.T. Hemann acknowledge funding from the MIT Center for Precision Cancer Medicine and the Ludwig Center at MIT.

## Author contributions

YL, AR, SH, and MTH designed the computational pipeline and experiments. YL implemented computational pipelines, preprocessed datasets, and performed analyses and experiments. NS performed nonlinear regression analyses. YL, NS, AR, SH, and MTH prepared the manuscript.

## Conflict of interest

The authors declare that they have no conflict of interest.

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
