## [Review Process File · Molecular Systems Biology]

Integrated regulatory models for inference of subtype-specific susceptibilities in glioblastoma

Yunpeng Liu, Ning Shi, Aviv Regev, and Shan He, and Michael Hemann
DOI: [10.15252/msb.20209506](https://doi.org/10.15252/msb.20209506)

Corresponding author: Michael Hemann (hemann@mit.edu)

Review Timeline:

Submission Date:	21st Feb 20
Editorial Decision:	9th Apr 20
Revision Received:	29th May 20
Editorial Decision:	24th Jul 20
Appeal Received:	27th Jul 20
Editorial Decision:	3rd Aug 20
Revision Received:	19th Aug 20
Editorial Decision:	24th Aug 20
Revision Received:	25th Aug 20
Accepted:	27th Aug 20

Editor: Maria Polychronidou

Transaction Report:

RE: Manuscript MSB-20-9506, "Integrated regulatory models for inference of subtype-specific susceptibilities in glioblastoma"

Thank you again for submitting your work to Molecular Systems Biology. We have now heard back from the three referees who agreed to evaluate your manuscript. As you will see below, the reviewers raise substantial concerns on your work, which unfortunately preclude its publication in Molecular Systems Biology.

The reviewers appreciate that inTRINSIC potentially represents a useful tool for future analyses. However, they point out that as it stands the study seems rather preliminary and they are not convinced that the advantages and clinical relevance of inTRINSIC and the related findings are well supported. The reviewers rated the conclusiveness of the study as Medium/Low and are not supportive of publication. As such, at this point we see no choice but to return the manuscript with the message that we cannot offer to publish it.

Nevertheless, as the reviewers did acknowledge that the approach is potentially relevant, we would not be opposed to considering a substantially revised and extended manuscript based on this work, provided that the issues raised by the reviewers can be convincingly addressed. Some of the more essential issues that would need to be addressed include:

- More recent datasets that more accurately reflect GBM subtypes need to be used (see related comments of reviewer #1 and #2).
- As reviewer #1 mentions, the network should be compared to a network built using an independent dataset.
- A comparison to existing approaches is warranted in order to clearly demonstrate the superiority and advantages of inTRINSIC.
- The clinical/translational relevance of the proposed approach (e.g. applicability to tumor heterogeneity data derived from single-cell analyses, relevance for therapeutic development etc.) needs to be better clarified and supported.
- Reviewers #1 and #2 point out that a major limitation at this point is the lack of follow up validations of the model predictions. Inclusion of such validations would significantly enhance the impact of the study. We understand that this requires a significant investment and may prove challenging and we would be happy to discuss with you what type of analyses would be feasible to perform in this direction.

All three reviewers provide constructive suggestions on how to address the points above and improve the study. We recognize that thoroughly addressing the referees' concerns would involve substantial further analyses with unclear outcome and we understand if in light of the substantial revisions required, you prefer to submit your study elsewhere. As mentioned above, if you are interested in a re-submission, I would be happy to look at a preliminary point by point response delineating how the issues raised can be addressed, so that we can work together on how to move forward.

A resubmitted work would have a new number and receipt date. It will be editorially evaluated afresh and its novelty will be re-assessed at the time of submission. As you probably understand, we can give no guarantee about its eventual acceptability. If you do decide to follow this course then we would ask you to enclose with your re-submission a point-by-point response to the points raised in the present review.

I am sorry that the review of your work did not result in a more favorable outcome on this occasion, but I hope that you will not be discouraged from sending your work to Molecular Systems Biology in the future. In any case, thank you for the opportunity to examine this work.

Reviewer #1:

Summary

Liu and colleagues describe an in silico approach termed 'integrative modeling of transcription regulatory interactions for systemic interference of susceptibility in cancer (inTRINSiC). The authors focus on describing transcription factor activity and their interplay in the malignant brain cancer glioblastoma, that can be classified into 4 subtypes (classical, mesenchymal, proneural, neural) by transcriptional/mutational profiling (as published by Verhaak et al., Cancer Cell, 2010). The presented results show overlap as well as differential regulatory activities of transcription factors according to glioblastoma subtype classification, which may causally link this activity to tumor subtype formation, also potentially leading to computational identification of glioblastoma molecular vulnerabilities (in a subtype-specific manner). MYBL2 is presented as an example of a potential 'drug target' as based on its regulatory role identified in proneural glioblastoma.

General remarks

While computationally-identifying regulatory transcription factor networks fueling cancer growth is not novel per se, the inTRINSiC approach entails several innovative elements with regards to integrating gene and protein-based databases, non-linear regression analysis, and vector-based algorithms. The methodology appears sound from the perspective of an experimental biologist (I cannot comment on the mathematical aspects). The paper is of interest for the brain and computational cancer research communities. While I am convinced that the presented computational modelling provides a good basis for further development and validation, my major concerns revolve around the versatility, timeliness, and potential translational aspects of the described in silico approach.

Major points

- The study appears to be based on gene expression data of The Cancer Genome Atlas Program (TCGA) of the year 2014. Can the authors clarify the relevance of the accessed data six years later (what changed since, is an update required)?
- How much overlap/difference can be found in the activities and networks of subtype-specific transcription factors in an independent data set such as the Chinese Glioma Genome Atlas (CGGA). A Venn diagram of this comparison should be provided and discussed.
- The combined focus on 'Verhaak subtypes' and 'bulk' gene expression data can be a limitation in terms of translational relevance. In particular, the current study does not take recent glioblastoma gene expression data at the single level into account. Key studies from the past 3 years should be cited (e.g., Darmanis et al., Cell Rep. 2017, Neftel et al., Cell, 2019) and the results should be discussed in the context of this recent literature. Along this line, the model would gain in significance if cellular plasticity at the single cell level as observed in patient tumors (Neftel et al., 2019; note the presence of mesenchymal, neural, astrocyte, and oligodendrocyte progenitor-like cell states) could be linked to transcription factor activity revealed by inTRINSiC.
- Functional characterization, for example based on MYBL1B loss-of-function and its effect on the proneural (versus other) signatures in patient-derived glioblastoma cell models would be required to strengthen the claim of identifying a potential proneural glioblastoma drug target. Without employing an experimental-computational cross-validation strategy, the authors should consider 'toning down' their claim of identifying 'drug targets' and current limitations of their approach (including modelling assumptions) should be more clearly highlighted in the discussion (an adequate limitation paragraph is currently lacking).

Minor points

- Why is the classical subtype flagged as an exception in terms of negative feedback mechanisms (changing regulatory behavior of transcription factors)? Wouldn't these feedback loops be expected independent of subtype profile? Can a sound explanation be provided?
- Why are immunotherapies (including Car T cells) described in the introduction - they are of no further importance for the narrative?
- Sottoriva et al., PNAS, 2013 should be cited for (first) identifying mixed glioblastoma subtype profiles in patient tumors (in addition to Patel et al., Science, 2014).
- The statement 'see materials and methods for details' should be removed and the key methodological concepts should be sufficiently and adequately explained throughout the narrative (ideally in a language that is also accessible for non-experts).
- Results: what is the context of the 'brain tumor cell line' data - a citation and introduction of the database (CCLE?) appears to be lacking?
- Page 3: replace 'logarithm space' with 'logarithmic space'.
- Page 6: correct 'mechanisms etc.33.'
- Figure 1: please introduce the sMAPE acronym.
- Figure 3: suggest to replace 'unique regulatory profiles with 'unique transcription factor regulatory

gene expression profiles' in the figure title.

Reviewer #2:

Manuscript Review

Date: 22 Mar 2020

Manuscript Number: MSB-20-9506

Manuscript Title: Integrated regulatory models for inference of subtype-specific susceptibilities in glioblastoma

In this manuscript the authors develop a computational method that infers transcriptional regulatory networks, which they refer to as Integrative Modeling of Transcription Regulatory Interactions for Systemic Inference of Susceptibility in Cancer (InTRINSiC). By developing this method, the authors aim to clarify the underlying regulatory mechanisms associated with the distinct molecular GBM subtypes, i.e., proneural (PN), classical (CL), neural (NE), and mesenchymal (MES). InTRINSiC utilizes a non-linear, thermodynamically-based regression to identify the TF-target gene relationships. The authors create a "basal" transcriptional regulatory network model that includes candidate TF-target gene pairs. Using this method, the authors analyze bulk-level omic-scale data of GBM samples (from TCGA) and identified distinct transcriptional regulatory network architecture associated with each GBM subtype.

In addition, the authors then connect this transcriptional network with a concomitantly designed protein signaling network which they use to infer what effects gene expression perturbations have at the protein signaling level by incorporating a predictive algorithm, based on an exponential ranking algorithm, which allows the user to predict how modulating TF expression will alter protein-level network states. The authors subsequently use this capability to identify "essential" TFs by simulating information flow (gene expression to protein signaling state). From this analysis, they were able to identify essential TFs like NFE2 and MYBL2, which were consistent with previously generated experimental data in CCLE and DepMap.

The authors have developed an interesting method that provides users with a tool to investigate the gene- to protein-level network architecture of a tumor as well as test and predict how coarse-grained perturbations of various TFs would affect the expression-phenotype of a tumor. The motivation of understanding the regulatory mechanisms associated with tumor heterogeneity, particularly in GBM, is indeed a highly relevant issue as this impedes treatment of the tumor. While the authors present a potentially useful tool, there are significant concerns related to the narrow context in which this tool is presented and the complete lack of experimental validations of model predictions.

Major concerns/comments:

- The authors use the original classification of GBM tumors defined by Verhaak et al. (2010). However, more recently, the four molecular subtypes was reduced down to three, where the original "neural" subtype was found to be an artifact of sampling of the leading edge of the tumor, which included non-malignant brain tissue (Wang et al. 2017). I'm concerned that the transcriptional network architecture results may change using the revised subtype annotation.

- I am not fully convinced that the claims made by the author that their results "explain gene expression variation in GBM" ("Transcription factor repurposing may be responsible for shaping subtype-specific transcriptomes" section). To a certain degree, yes, these results characterize distinct, bulk-level expression differences between subtypes, but there are other factors contributing to gene expression variability. One primary factor driving expression variability is the stochastic nature of gene expression, which is reflected at the single-cell level. The results presented infer distinct TF-gene and TF-protein interactions that distinguish GBM subtypes, which is one level of variation. However, because there are multiple levels of variation and heterogeneity in biology, the earlier statement is only somewhat substantiated. Perhaps a clarification in that the variation being characterized by the authors' analysis is at the level of GBM subtypes, would help clarify this claim.

- Multiple regulatory network inference algorithms have been developed and have been used to identify networks within the context of GBM, e.g. Plaisier et al. Cell Systems 2016 (which was surprisingly not cited in this manuscript). In this example, the authors incorporated protein-, mRNA, miRNA, and mutational data to construct a multi-omic scale transcriptional regulatory network, encompassing a larger scope of data modalities and genomic features than what is presented here. Here, the authors focus on gene-expression and protein-level characterization, which are important aspects governing transcriptional state. However, mutational profiles certainly impact the transcriptional heterogeneity pervasive in GBM tumors, which is something that is not addressed by the authors. As the authors are motivated to understand the regulatory mechanisms underlying GBM tumor heterogeneity, I am surprised why such information (mutational profiling of some kind) is not considered at all.

- Another major concern has to do with the complete lack of experimental validations. The authors look for consistent patterns of transcriptional changes upon knockdown of relevant TFs in the CCLE. While this analysis does provide supporting evidence that the regulatory interactions are likely to be functional they provide no insight into the biological and mechanistic relevance of the regulatory interactions on cancer phenotypes. Thus, it is not surprising the final sentence of the results section is a conjecture "...suggesting that subtype-specific TF interactions may be key to maintaining cell state and viability in the corresponding subtype." In absence of such validations this manuscript is much better suited for a specialist journal with a focus on computational biology.

- In a related comment, it would help to add some text that would help distinguish how this method differs from others. For example - the predictive capabilities of this methodology are very interesting and offer a tool that would potentially help to narrow in on specific targets. I'm not sure if this capability is something that many of the pre-existing network inference algorithms have. Mentioning this distinction would strengthen the claim of the utility of this algorithm.

- Given the predictive capabilities of the model, have the authors considered how they might use this tool to identify TFs that may cause a tumor to transition from one subtype to another? Clinically, GBMs of a proneural or classical subtype often recur as a mesenchymal subtype. It would be interesting to see if through their exponential ranking-based algorithm if the authors can identify what TF(s) can cause a proneural subtype to transition to a mesenchymal subtype. This type of analysis would add meaningful substance to this work as plasticity of tumor cells, particularly in GBM, is another related challenge in treating cancers.

- The use of the basal CNS TF-gene network raises some concern/questions. As this highlights the

strongest TF-gene connections, it averages away any of the distinct cell-type and region-specific (amongst other critical distinguishing features of the highly differentiated brain) TF-gene relationships. Because the cell-type of origin remains an open question in the field, it is somewhat surprising that a "mass-averaged" basal TF-gene network works in inferring distinct network structures for something as dysregulated as GBM. Further analysis is required in addition to in depth discussion as to why using a mass-averaged basal network in describing such a heterogeneous tumor like GBM works, as opposed to inferring a purely data-driven network.

- Clarification on heterogeneity levels: bulk vs single-cell and how bulk tumor heterogeneity should be described at the level of cell population structure, proportions of various cell types and tumor cells of a particular subtype. The authors discuss this issue of heterogeneity at bulk and single-cell levels, but the results highlight the inferred network derived from bulk data, which represent an amalgamation of distinct cell-types and subpopulations of PN, MES, CL tumor cells in different proportions. While interesting, this type of algorithm would be more effective (clinically) if it were able to identify distinct regulatory mechanisms that are applicable to single-cell level subpopulations of tumors.

Minor comments:

Figure 1, panels e through h are illegible and, therefore, difficult to interpret

Page 7, 2nd paragraph - "We then trained an elastic net regression model for each perturbed TF to find a linear combination of proteins [,] the changes of which best predict the TF's gene essentiality..."

Page 11, "Inferring protein activity inferred from an integrated transcription regulation-signaling network" - "To infer protein signaling activity, we used ... similar to PageRank algorithm47 [ref]."

Reviewer #3:

Summary

GBM is a deadly malignancy that lacks effective treatment. Despite previous work that stratifies the disease into four subtypes, the regulatory changes that underlie these differences are unclear. The authors develop a new algorithm, inTRINSIC, that incorporates regulatory information from multiple sources to build a network of transcriptional regulation in a subtype-specific manner. These networks are parametrized with a non-linear regression, then coupled with a network of protein signaling based on known protein-protein interactions. These elements combine to form a multilayer network that allows for in silico perturbation of transcription factor (TF) expression, opening the door for investigations of patient data without requiring arduous or infeasible experiments.

This novel method was leveraged to analyze GBM samples within the existing Verhaak subtypes. inTRINSIC identified sets of TFs with differential regulatory activity in each GBM subtype, including both novel and known markers. The MYBL2 gene is highlighted as a regulatory TF for the Proneural subtype; a gene that has both appeared in previous analyses and been shown to interact with ASCL1, another known regulatory TF of the Proneural subtype, in vitro.

More interesting than the GBM-specific conclusions is the methodology, however. The inTRINSIC algorithm can be broken down into five steps:

1 - Inference of a regulatory network from the union of two sources; existing networks from the FANTOM5 database and a novel network based on open-chromatin and TF binding motif data. For the latter network, chromatin information came from DNase hypersensitivity sequence (DHS-seq) datasets from ENCODE, while motif information came from the JASPAR database.

2 - The network from step 1 serves as a scaffold for nonlinear regressions. The expression of each target is modeled as the independent sums of the effect of saturable TFs. The regression will assign both a sign and a strength to each regulatory interaction.

3 - A protein activity network is generated by combining databases of known protein-protein interactions; SPRING, which provides signed links between proteins; and FunCoup, which provides likelihoods on each interaction. The activity of the proteins in the resulting network are then inferred with the Exponential Ranking algorithm, with $p(0)$ (the initial protein activity) given by the expression of each protein scaled to sum to one.

4 - Steps 1-3 create a multi-layer network; changes in TF gene expression drive changes in target expression, and the total vector of expression is fed into the protein activity network to analyze how signaling activity changes based on perturbations to each TF.

5 - An elastic-net is trained to predict essentiality scores from downstream differential activity resulting from in silico perturbation of TF expression on CCLE cell lines. These regression are then used to predict gene essentiality scores from similar perturbations carried out in patient data.

General Remarks

The inTRINSIC tool developed in this paper potentially provides an incredible tool to researchers; the ability to perturb transcriptional profiles in silico. Avoiding intractable, expensive, and/or time consuming knock down experiments typically necessary to understand the full impact of a TF would have a huge impact on our ability to parse out causative regulatory information. To speak colloquially for a moment, the ability to adjust one lever of the transcriptomic architecture and model how the rest will adjust is, simply put, cool.

That said, the actual functional implications of this technique are a bit more nebulous. The authors utilize these perturbations to predict gene essentiality scores as a method of identifying regulatory TFs that define GBM subtypes based on differential activity. However, the VIPER algorithm] already permits similar analyses and has been used to predict master regulators (MRs) that control cell state across a variety of malignancies. It is unclear whether inTRINSIC provides a functional improvement over existing methodology in terms of identifying these MRs, and the authors do not perform any comparisons to identify the differences between them.

Going further, the ability of this methodology to cut down on the complexity of large scale perturbational assays is not clear either. For instance, TFs are typically not directly targetable by drugs, and predicting which drugs will affect which TFs in a given context typically requires a perturbational assay in a sample as cell line or tissue as close to the sample of interest as possible. As such, while this method should be able to predict how a given sample's transcriptional architecture will change if a TF is knocked down, bridging the gap to actionable clinical information

doesn't seem possible. To be clear, the author's have made no such claims, but this avenue was one of the immediate possible implications that came to mind upon reading the paper that unfortunately does not seem viable upon further scrutiny.

Major Points

In the first step of the inTRINSIC pipeline, previously existing regulatory networks are combined with both open chromatin data and known motif-binding maps. The first half of that formula - the pre-existing networks - come from brain tissues in the FANTOM5 project. While these networks are no doubt more relevant to the GBM samples discussed, there are still major transcriptomic differences between healthy and cancerous tissue. Meanwhile, the open chromatin data (from ENCODE) and the motif-binding information (from JASPAR) suffer from similar problems of context specificity, though the cell lines in question are likely a better model for the GBM tissue being studied than healthy tissue.

However, this approach based on pre-existing data for TFs with known binding motifs opens up an obvious hole in the methodology; no proteins with unknown binding motifs can be studied. The final transcriptomic network contains 518 TFs, but the human genome contains roughly 1800 TFs and an additional 600 co-TFs that all can play major roles as checkpoint modulators. Finally, the method of mapping from motif-binding graphs to a regulator-target graph - based on the nearest gene method often employed in ATACseq or other similar techniques - is prone to even more issues related to context specificity, as well as an inability to resolve more complicated regulatory structures in the chromatin itself].

All of these issues can be boiled down to one core issue; relying almost entirely on the literature for network construction is problematic due to the various bias associated with the construction of this prior knowledge. These biases are unavoidable with the method presented in the paper, but comparisons with more unbiased methods of network reconstruction are warranted. Various correlation based network reconstruction algorithms, machine learning based methods (SCENIC), or mutual information based (ARACNe)] are all publicly available and could be used to infer networks for comparison to the method proposed in the paper. It's difficult to go so far as to say which method - biased or unbiased - is superior in a vacuum, but the authors should at least acknowledge some of the limitations inherent in their approach and address them in some manner.

As a final note, the author's critiques of existing methods for regulatory network generation are not entirely accurate. ARACNe, for instance, has a pair of parameters that capture both the relative certainty of an edge's existence and the directionality of its regulation. Taken together - along with ARACNe's extremely low false positive rate - these two parameters essentially capture the relative strength of the TF-target interaction. Similar to the previous paragraph, flaws in the manuscript's critique of other network generation methods could be resolved via a more direct comparison between them and a deeper discussion of the pros and cons of this novel method.

Moving past the generation of the network framework, the use of non-linear regression to parametrize regulatory edges is interesting and appropriate. A brief comment discussing why a method that included cross terms for co-regulatory effects was either not possible or not attempted would be relevant here. The subsequent method of determining the correlation of regulators in order to determine co-regulatory proteins also seems robust in theory. However, the use of the RANSAC algorithm - a technique designed more to leave out outliers due to poor data collection or signal loss - seems inappropriate here, as the vectors being correlated are not raw data but an inferred statistic. If RANSAC was implemented because methods based purely on

correlation were not sufficient, that could indicate a problem with overfitting in this context.

Minor Points

Beyond the more significant challenges listed to the methodology discussed above, there are a few areas where explanations could be more clear, or where there has simply been an editing oversight. These are listed in bulleted form below:

The explanation in the Results section about subtype-specific regulatory profiles, particularly the explanation about regulatory signatures versus co-regulatory signatures, is a bit unclear. A reorganization that first explains the regulatory signature, then the co-regulatory signature, then the comparison between them - perhaps in different sections - might be easier to interpret.

The reference for the PageRank algorithm is missing on page 11.

Alvarez MJ, Shen Y, Giorgi FM, et al. Functional characterization of somatic mutations in cancer using network-based inference of protein activity. *Nat Genet.* 2016;48(8):838-847. doi:10.1038/ng.3593

Aibar S, González-Blas CB, Moerman T, et al. SCENIC: single-cell regulatory network inference and clustering. *Nat Methods.* 2017;14(11):1083-1086. doi:10.1038/nmeth.4463

Lachmann A, Giorgi FM, Lopez G, Califano A. ARACNe-AP: gene network reverse engineering through adaptive partitioning inference of mutual information. *Bioinformatics.* 2016;32(14):2233-2235. doi:10.1093/bioinformatics/btw216

Response to Reviewers –

We are truly indebted to the reviewers for their thoughtful examination of this manuscript. Indeed, we believe that the revised manuscript has been considerably improved based on the insightful suggestions made during the initial review. In this Response to Reviewer section, we first summarize the major changes that are present in this revised manuscript. Subsequently, we provide a more detailed point-by-point response to the reviewers. Again, we would like to thank the reviewers for the time and effort made during their examination of this work.

Major changes -

1. We have reassessed our subtype classification reflecting the discovery by Wang et al. (2017) that the Neural subtype may be an artifact of normal tissues surrounding tumors as well as cell type mixtures. Here, “Neural” samples get reassigned to the other 3 labels in the new classification scheme (Supplementary Figure 1b). Importantly, the addition of these new samples to each subtype did not significantly skew our results (Supplementary Figure 1d). Indeed, our data is consistent with that found in the Wang et al study, and we believe that omission of the “Neural” samples in the TCGA dataset most likely provides the best alternative. The Neural samples, while displaying a unique expression signature, are unlikely to provide consistent insight on subtype-specific regulatory programs due to cell state mixing effects. Additionally, we did not find significant updates to the TCGA expression dataset we used in our initial analysis.
2. We have constructed a new set of network models using the Chinese Glioma Genome Atlas (CGGA) GBM dataset consisting of 172 Classical, Proneural and Mesenchymal subtypes, and compared network parameters with those from the TCGA samples. We found significant concordance between the two (Figure 1, Supplementary Figure 1).
3. We have provided a systematic comparison of the network edges obtained from ARACNe and that of our inTRINSiC pipeline (Supplementary Figure 2b-d). While ARACNe is clearly a powerful analysis tool, we found that our method achieves a broader range of TF-TF interaction discovery while maintaining the same accuracy at capturing known BIOGRID interactions as ARACNe. We now include a discussion of advantages/disadvantages of inTRINSiC compared to ARACNe. We have also added discussion on unique features of inTRINSiC compared to other pipelines like SCENIC and SYGNAL in our Introduction and Discussion sections.
4. We have included a comparative analysis of recent single cell expression-based studies examining multiple layers of intratumoral heterogeneity in GBM. Specifically, we have compared our discoveries with those from single cell studies (e.g. DDIT3 as a key Mesenchymal factor in both our analyses and Neftel et al. (2019)) and further demonstrated that our models, built on bulk expression data, are capable of generating verifiable insights into the regulatory heterogeneity of GBM. Additionally, we have implemented a plasticity analysis where we inspect how GBM transcriptomes shift in

response to transcription factor perturbation. Here, one of the exciting and intriguing observations that emerges is the relevance of STAT3, a Mesenchymal signature TF, whose *in silico* knockdown induced a substantial shift of Mesenchymal samples towards the Classical subtype cluster (Figure 3e, Supplementary Figure 2e).

5. Despite the current infeasibility of wet bench experimental validation, we now present a systematic and comprehensive analysis of the DepMap dependency data in Figure 5. This allowed us to bring an extensive set of experimental validation to bear on our analyses. In addition to scoring each transcription factor based on the concordance between their predicted essentiality and DepMap experimentally determined scores, we have also filtered for those TFs that show similar subtype-specificity in their predicted essentiality scores as that seen in DepMap scores. Notably, for the TF that scores among the top in all of the above criteria, MYBL2, we demonstrate that its expression can significantly segregate patient survival in both the TCGA and CGGA datasets in a Proneural subtype-specific way, suggesting that it indeed represents a potential drug target in this GBM subtype.

Reviewer #1:

Summary

Liu and colleagues describe an *in silico* approach termed 'integrative modeling of transcription regulatory interactions for systemic interference of susceptibility in cancer (inTRINSiC). The authors focus on describing transcription factor activity and their interplay in the malignant brain cancer glioblastoma, that can be classified into 4 subtypes (classical, mesenchymal, proneural, neural) by transcriptional/mutational profiling (as published by Verhaak et al., Cancer Cell, 2010). The presented results show overlap as well as differential regulatory activities of transcription factors according to glioblastoma subtype classification, which may causally link this activity to tumor subtype formation, also potentially leading to computational identification of glioblastoma molecular vulnerabilities (in a subtype-specific manner). MYBL2 is presented as an example of a potential 'drug target' as based on its regulatory role identified in proneural glioblastoma.

General remarks

While computationally-identifying regulatory transcription factor networks fueling cancer growth is not novel per se, the inTRINSiC approach entails several innovative elements with regards to integrating gene and protein-based databases, non-linear regression analysis, and vector-based algorithms. The methodology appears sound from the perspective of an experimental biologist (I cannot comment on the mathematical aspects). The paper is of interest for the brain and computational cancer research communities. While I am convinced that the presented computational modelling provides a good basis for further development and validation, my major

concerns revolve around the versatility, timeliness, and potential translational aspects of the described in silico approach.

We thank the reviewer for their comments regarding the quality and innovative potential of this work. We believe that our revisions in response speak directly to concerns regarding versatility, timeliness and translational potential our approach.

Major points

- The study appears to be based on gene expression data of The Cancer Genome Atlas Program (TCGA) of the year 2014. Can the authors clarify the relevance of the accessed data six years later (what changed since, is an update required)?

We thank the reviewer for pointing this out. Upon further investigation of this database, TCGA has not added new samples since 2013. Our analysis used all of the samples in the repository that had microarray expression data (544 samples, 407 excluding the spurious Neural subtype), and these samples still constitute the largest well-annotated set of glioblastoma patient expression data.

- How much overlap/difference can be found in the activities and networks of subtype-specific transcription factors in an independent data set such as the Chinese Glioma Genome Atlas (CGGA). A Venn diagram of this comparison should be provided and discussed.

This is an important suggestion, and we have tried to find the best possible approach to effectively use the CGGA database. Since the CGGA database has about half (249) the number of TCGA GBM samples, it creates overfitting issues due to relatively small numbers of samples per subtype (~70 compared to more than 120 for TCGA). To most effectively use the CGGA dataset as an independent validation of our findings, we performed survival analysis of MYBL2 scores in CGGA patients (Figure 5g). Here, we found a consistent, significant difference in survival between MYBL2-high and MYBL2-low patients. We believe that this result provides validation for the translational potential of our approach.

- The combined focus on 'Verhaak subtypes' and 'bulk' gene expression data can be a limitation in terms of translational relevance. In particular, the current study does not take recent glioblastoma gene expression data at the single level into account. Key studies from the past 3 years should be cited (e.g., Darmanis et al., Cell Rep. 2017, Neftel et al., Cell, 2019) and the results should be discussed in the context of this recent literature. Along this line, the model would gain in significance if cellular plasticity at the single cell level as observed in patient tumors (Neftel et al., 2019; note the presence of mesenchymal, neural, astrocyte, and oligodendrocyte progenitor-like cell states) could be linked to transcription factor activity revealed by inTRINSiC.

In the revised manuscript, we now cite and discuss the above mentioned work in multiple sections of our revised manuscript. In our analysis, we built our models on bulk expression data

due to its broad coverage of gene expression in large patient cohorts. However, when we compare bulk expression data with single cell analyses, we see significant concordance of our models with models built on single-cell expression data (revised Figure 1d). Indeed, this issue was of central importance to our analysis, given our interest (the Regev lab) in single cell data.

Plasticity is a very interesting point, and we thank the reviewer for raising this issue. We performed systematic perturbation of 515 TFs and report the relative shifts in transcriptomes in TCGA patients (Figure 3e, Supplementary Figure 2e and Supplementary Table 1). Notably, we found that perturbing Mesenchymal 'master regulators' such as STAT3 *in silico* results in a shift of a substantial number of Mesenchymal samples towards the Classical subtype cluster (Figure 3e). We believe that this result has significant relevance regarding the evolution of resistance towards subtype-specific therapies or the ability to move therapy-resistant subtypes towards a more responsive state. Notably, we also believe that this data speaks to the versatility of our approach.

- Functional characterization, for example based on MYBL1B loss-of-function and its effect on the proneural (versus other) signatures in patient-derived glioblastoma cell models would be required to strengthen the claim of identifying a potential proneural glioblastoma drug target. Without employing an experimental-computational cross-validation strategy, the authors should consider 'toning down' their claim of identifying 'drug targets' and current limitations of their approach (including modelling assumptions) should be more clearly highlighted in the discussion (an adequate limitation paragraph is currently lacking).

We thank the reviewer for this really important comment. We agree that the limitations in our model predictions need to be addressed in more detail. We highlight transcription factors that we predicted to show subtype-specific essentiality as potential 'drug targets' due to the apparent selective *in vitro* survival disadvantage conferred by loss of that factor in cell lines classified as a particular subtype, and this does not necessarily mean that loss of the factor will also elicit the same specific effects in patient samples *in vivo*. However, since we also observe that our model (built on *in vitro* cell line expression data) predicts similar selective disadvantages when applied to patient expression profiles and see clear segregation of survival for our top hit MYBL2 in patients, we gain some confidence in prioritizing these factors for future *in vivo* validation studies. This point is now emphasized, as are the limitations of our analysis, in the discussion.

Minor points

- Why is the classical subtype flagged as an exception in terms of negative feedback mechanisms (changing regulatory behavior of transcription factors)? Wouldn't these feedback loops be expected independent of subtype profile? Can a sound explanation be provided?

We concede that negative feedback loops are indeed expected to operate independent of GBM subtype. However, in this particular analysis where we inspect the association between transcription factor and target gene expression, we speculate that the opposite changes we see in the Classical subtype compared to that in the other three may be due to a *subset of subtype-specific* negative feedback loops that only operate in the Classical subtype.

- Why are immunotherapies (including Car T cells) described in the introduction - they are of no further importance for the narrative?

We apologize for the omission of a follow-up discussion on immunotherapy. In this revised manuscript (Page 10), we now implicate factors such as STAT3 (a signature TF discovered by inTRINSiC) in GBM. This gene has been shown to play key roles in creating an immunosuppressive tumor microenvironment. This data, as well as a discussion of its relevance to immunotherapy, is now provided in this submission. Indeed, given the current relevance of immunotherapy, we believe that the identification of STAT3 here represents a very timely application of inTRINSiC towards immune-oncology.

- Sottoriva et al., PNAS, 2013 should be cited for (first) identifying mixed glioblastoma subtype profiles in patient tumors (in addition to Patel et al., Science, 2014).

Thank you for pointing this out. We have added this to our bibliography.

- The statement 'see materials and methods for details' should be removed and the key methodological concepts should be sufficiently and adequately explained throughout the narrative (ideally in a language that is also accessible for non-experts).

Thank you for the useful comment. We have clarified the description of technical terms throughout the main text for a broader audience.

- Results: what is the context of the 'brain tumor cell line' data - a citation and introduction of the database (CCLE?) appears to be lacking?

- Page 3: replace 'logarithm space' with 'logarithmic space'.

- Page 6: correct 'mechanisms etc.33.'

- Figure 1: please introduce the SMAPE acronym.

- Figure 3: suggest to replace 'unique regulatory profiles with 'unique transcription factor regulatory gene expression profiles' in the figure title.

We thank the reviewer for the above four points and have made all of the suggested changes/additions.

Reviewer #2:

Manuscript Review

Date: 22 Mar 2020

Manuscript Number: MSB-20-9506

Manuscript Title: Integrated regulatory models for inference of subtype-specific susceptibilities in glioblastoma

In this manuscript the authors develop a computational method that infers transcriptional regulatory networks, which they refer to as Integrative Modeling of Transcription Regulatory Interactions for Systemic Inference of Susceptibility in Cancer (InTRINSiC). By developing this method, the authors aim to clarify the underlying regulatory mechanisms associated with the distinct molecular GBM subtypes, i.e., proneural (PN), classical (CL), neural (NE), and mesenchymal (MES). InTRINSiC utilizes a non-linear, thermodynamically-based regression to identify the TF-target gene relationships. The authors create a "basal" transcriptional regulatory network model that includes candidate TF-target gene pairs. Using this method, the authors analyze bulk-level omic-scale data of GBM samples (from TCGA) and identified distinct transcriptional regulatory network architecture associated with each GBM subtype.

In addition, the authors then connect this transcriptional network with a concomitantly designed protein signaling network which they use to infer what effects gene expression perturbations have at the protein signaling level by incorporating a predictive algorithm, based on an exponential ranking algorithm, which allows the user to predict how modulating TF expression will alter protein-level network states. The authors subsequently use this capability to identify "essential" TFs by simulating information flow (gene expression to protein signaling state). From this analysis, they were able to identify essential TFs like NFE2 and MYBL2, which were consistent with previously generated experimental data in CCLE and DepMap.

The authors have developed an interesting method that provides users with a tool to investigate the gene- to protein-level network architecture of a tumor as well as test and predict how coarse-grained perturbations of various TFs would affect the expression-phenotype of a tumor. The motivation of understanding the regulatory mechanisms associated with tumor heterogeneity, particularly in GBM, is indeed a highly relevant issue as this impedes treatment of the tumor. While the authors present a potentially useful tool, there are significant concerns related to the narrow context in which this tool is presented and the complete lack of experimental validations of model predictions.

We thank the reviewer for their comments regarding the relevance and potential utility of our approach. In our revised manuscript, we broadened the scope of experimental application of this approach within GBM, as well as provide significant preclinical validation in CGGA and DepMap databases.

Major concerns/comments:

- The authors use the original classification of GBM tumors defined by Verhaak et al. (2010). However, more recently, the four molecular subtypes was reduced down to three, where the original "neural" subtype was found to be an artifact of sampling of the leading edge of the tumor, which included non-malignant brain tissue (Wang et al. 2017). I'm concerned that the transcriptional network architecture results may change using the revised subtype annotation.

We thank the reviewer for pointing out this important point. We carefully examined the revised subtype annotation for GBM samples in the revised manuscript and comprehensively reran the algorithm with the revised subtypes. Importantly, using this new annotation did not introduce any systematic bias or affect the F values for the three remaining "non-Neural" subtypes. Indeed, we can simply omit the Neural subtype, consistent with the currently accepted classification system. We believe that this data speaks to the potential robustness and versatility of this approach.

- I am not fully convinced that the claims made by the author that their results "explain gene expression variation in GBM" ("Transcription factor repurposing may be responsible for shaping subtype-specific transcriptomes" section). To a certain degree, yes, these results characterize distinct, bulk-level expression differences between subtypes, but there are other factors contributing to gene expression variability. One primary factor driving expression variability is the stochastic nature of gene expression, which is reflected at the single-cell level. The results presented infer distinct TF-gene and TF-protein interactions that distinguish GBM subtypes, which is one level of variation. However, because there are multiple levels of variation and heterogeneity in biology, the earlier statement is only somewhat substantiated. Perhaps a clarification in that the variation being characterized by the authors' analysis is at the level of GBM subtypes, would help clarify this claim.

We appreciate the reviewer's useful comment. We fully agree that the differential TF regulatory profiles identified through our models only explain a small part of gene expression variation, and that we are capturing specific *subtype-specific* variations here. Additionally, since we are using a thermodynamic steady-state model to describe gene expression, our model currently does not take stochasticity in gene expression into account. We do like to reiterate, however, that since we integrated open chromatin data to infer candidate TF-target gene pairs, our model has taken multiple layers of regulation (TF-TF coregulation, chromatin remodeling etc.) into consideration, which are collapsed into the F-values inferred for each TF-gene pair. This issue is now clarified more extensively in the discussion.

- Multiple regulatory network inference algorithms have been developed and have been used to identify networks within the context of GBM, e.g. Plaisier et al. Cell Systems 2016 (which was surprisingly not cited in this manuscript). In this example, the authors incorporated protein-, mRNA, miRNA, and mutational data to construct a multi-omic scale transcriptional regulatory network, encompassing a larger scope of data modalities and genomic features than what is presented here. Here, the authors focus on gene-expression and protein-level characterization, which are important aspects governing transcriptional state. However, mutational profiles certainly impact the transcriptional heterogeneity pervasive in GBM tumors, which is something that is not addressed by the authors. As the authors are motivated to understand the regulatory

mechanisms underlying GBM tumor heterogeneity, I am surprised why such information (mutational profiling of some kind) is not considered at all.

We are indeed aware of this important study in Cell Systems and have expanded our discussion to include a comparison between the approach used in this study and that used in ours. We did not explicitly compare our study with that of Plaisier et al. because we recognize that we are in fact tackling the problem of understanding GBM heterogeneity from a vastly different angle. The SYGNAL pipeline introduced by Plaisier et al. focuses largely on genes with somatic mutations and asks how these alterations affect downstream regulation, and candidate TFs were filtered using *known* associations with GBM. In our inTRINSiC pipeline, we start with as few constraints/filtering on the input data as possible to decrease bias by prior knowledge, and use a largely data-driven approach to allow our model to discover GBM sample-specific regulatory edges. These two approaches complement each other in many aspects. In our models, we did not explicitly model the effects of genetic alterations for two reasons: 1) lack of knowledge of how certain mutations (which are also often sparse across samples) would affect gene function, and 2) explicit modeling of TF 'regulatory activity' as an abstract concept that encompasses multiple layers of regulation is expected to take the effects of mutation, if functionally relevant, into account. This seems to be the case in our analyses and we explicitly explore this important issue in our updated Discussion section.

- Another major concern has to do with the complete lack of experimental validations. The authors look for consistent patterns of transcriptional changes upon knockdown of relevant TFs in the CCLE. While this analysis does provide supporting evidence that the regulatory interactions are likely to be functional they provide no insight into the biological and mechanistic relevance of the regulatory interactions on cancer phenotypes. Thus, it is not surprising the final sentence of the results section is a conjecture "...suggesting that subtype-specific TF interactions may be key to maintaining cell state and viability in the corresponding subtype." In absence of such validations this manuscript is much better suited for a specialist journal with a focus on computational biology.

We appreciate the reviewer's comment. While we have a limited ability to provide wet bench validation due to current circumstances, we now provide a more comprehensive analysis of how our predictions compare with DepMap data. Here, the TF that scores among the top in all of the above criteria, MYBL2, is a clear Proneural dependency. Additionally, we now demonstrate that MYBL2's expression can significantly segregate patient survival in both the TCGA and CGGA datasets in a Proneural subtype-specific way, suggesting that it indeed represents a potential drug target in this GBM subtype. We believe that these analyses provide important preclinical validation of our approach.

- In a related comment, it would help to add some text that would help distinguish how this method differs from others. For example - the predictive capabilities of this methodology are very interesting and offer a tool that would potentially help to narrow in on specific targets. I'm not sure if this capability is something that many of the pre-existing network inference algorithms have. Mentioning this distinction would strengthen the claim of the utility of this algorithm.

We thank the reviewer for this constructive comment. Indeed, important inference algorithms including ARACNe exist to identify genes important for specific biological contexts. However, our pipeline provides information on the magnitude of regulation (i.e. magnitude of log₂ F-values) in addition to directionality. We now included comparison of these approaches (Supplementary Figure 2) and additional discussion of related methods and unique features of our methodology in the revised text.

- Given the predictive capabilities of the model, have the authors considered how they might use this tool to identify TFs that may cause a tumor to transition from one subtype to another? Clinically, GBMs of a proneural or classical subtype often recur as a mesenchymal subtype. It would be interesting to see if through their exponential ranking-based algorithm if the authors can identify what TF(s) can cause a proneural subtype to transition to a mesenchymal subtype. This type of analysis would add meaningful substance to this work as plasticity of tumor cells, particularly in GBM, is another related challenge in treating cancers.

We thank the reviewer for this very interesting point. We have now extended our analyses to understanding phenotypic plasticity in GBM, and we are excited to present the results in the revised manuscript. We performed systematic perturbation of 515 TFs and report the relative shifts in transcriptomes in TCGA patients (Figure 3e, Supplementary Figure 2e and Supplementary Table 1). Specifically, we found that perturbing Mesenchymal 'master regulators' such as STAT3 *in silico* results in a shift of a substantial number of Mesenchymal samples towards the Classical subtype cluster (Figure 3e).

- The use of the basal CNS TF-gene network raises some concern/questions. As this highlights the strongest TF-gene connections, it averages away any of the distinct cell-type and region-specific (amongst other critical distinguishing features of the highly differentiated brain) TF-gene relationships. Because the cell-type of origin remains an open question in the field, it is somewhat surprising that a "mass-averaged" basal TF-gene network works in inferring distinct network structures for something as dysregulated as GBM. Further analysis is required in addition to in depth discussion as to why using a mass-averaged basal network in describing such a heterogeneous tumor like GBM works, as opposed to inferring a purely data-driven network.

We appreciate the reviewer's comment present clarification regarding this concern in the revised manuscript. Indeed as one of our groups (the Regev lab) has done considerable work based on single cell analysis, it was very important for us to compare bulk and single cell approaches. While we do agree that using bulk expression data averages away effects of cell states in subpopulations, and that the cell-type of origin cannot be readily identified using analyses on bulk data, we argue that transcriptional heterogeneity, at least at a level that is also seen in single cell transcriptome profiling (as seen the 2014 study by Patel et al.), can be largely dissected using bulk expression profiles which capture genes with lower expression more effectively than single cell transcriptomics. In addition, now show a significant concordance

between the models we built from bulk and single-cell data with respect to the genes covered by both datasets (Figure 1d).

- Clarification on heterogeneity levels: bulk vs single-cell and how bulk tumor heterogeneity should be described at the level of cell population structure, proportions of various cell types and tumor cells of a particular subtype. The authors discuss this issue of heterogeneity at bulk and single-cell levels, but the results highlight the inferred network derived from bulk data, which represent an amalgamation of distinct cell-types and subpopulations of PN, MES, CL tumor cells in different proportions. While interesting, this type of algorithm would be more effective (clinically) if it were able to identify distinct regulatory mechanisms that are applicable to single-cell level subpopulations of tumors.

Again, we thank the reviewer for this important point. As mentioned in the previous comment, we would like to highlight the idea that the expression subtypes identified from bulk expression data may be a result of a single subpopulation of cells bearing the corresponding expression signature *at the single cell level* dominating the tumor sample, as seen from the analyses by Patel et al. (2014). In this sense, we believe that building our models upon bulk expression data with more comprehensive coverage of gene expression generates highly relevant insight. To address the reviewer's final concern, our modeling pipeline is readily applicable to single-cell expression data whenever high coverage data from multiple patients are available. Again, we believe that this speaks to the versatility of our approach.

Minor comments:

Figure 1, panels e through h are illegible and, therefore, difficult to interpret

Page 7, 2nd paragraph - "We then trained an elastic net regression model for each perturbed TF to find a linear combination of proteins [,] the changes of which best predict the TF's gene essentiality..."

Page 11, "Inferring protein activity inferred from an integrated transcription regulation-signaling network" - "To infer protein signaling activity, we used ... similar to PageRank algorithm⁴⁷ [ref]."

We thank the reviewer for pointing out the above issues and we have implemented these recommendations in our revised manuscript.

Reviewer #3:

Summary

GBM is a deadly malignancy that lacks effective treatment. Despite previous work that stratifies the disease into four subtypes, the regulatory changes that underlie these differences are unclear. The authors develop a new algorithm, inTRINSIC, that incorporates regulatory

information from multiple sources to build a network of transcriptional regulation in a subtype-specific manner. These networks are parametrized with a non-linear regression, then coupled with a network of protein signaling based on known protein-protein interactions. These elements combine to form a multilayer network that allows for in silico perturbation of transcription factor (TF) expression, opening the door for investigations of patient data without requiring arduous or infeasible experiments.

This novel method was leveraged to analyze GBM samples within the existing Verhaak subtypes. inTRINSIC identified sets of TFs with differential regulatory activity in each GBM subtype, including both novel and known markers. The MYBL2 gene is highlighted as a regulatory TF for the Proneural subtype; a gene that has both appeared in previous analyses and been shown to interact with ASCL1, another known regulatory TF of the Proneural subtype, in vitro.

More interesting than the GBM-specific conclusions is the methodology, however. The inTRINSIC algorithm can be broken down into five steps:

1 - Inference of a regulatory network from the union of two sources; existing networks from the FANTOM5 database and a novel network based on open-chromatin and TF binding motif data. For the latter network, chromatin information came from DNase hypersensitivity sequence (DHS-seq) datasets from ENCODE, while motif information came from the JASPAR database.

2 - The network from step 1 serves as a scaffold for nonlinear regressions. The expression of each target is modeled as the independent sums of the effect of saturable TFs. The regression will assign both a sign and a strength to each regulatory interaction.

3 - A protein activity network is generated by combining databases of known protein-protein interactions; SPRING, which provides signed links between proteins; and FunCoup, which provides likelihoods on each interaction. The activity of the proteins in the resulting network are then inferred with the Exponential Ranking algorithm, with $p(0)$ (the initial protein activity) given by the expression of each protein scaled to sum to one.

4 - Steps 1-3 create a multi-layer network; changes in TF gene expression drive changes in target expression, and the total vector of expression is fed into the protein activity network to analyze how signaling activity changes based on perturbations to each TF.

5 - An elastic-net is trained to predict essentiality scores from downstream differential activity resulting from in silico perturbation of TF expression on CCLE cell lines. These regression are then used to predict gene essentiality scores from similar perturbations carried out in patient data.

General Remarks

The inTRINISIC tool developed in this paper potentially provides an incredible tool to researchers; the ability to perturb transcriptional profiles in silico. Avoiding intractable, expensive, and/or time consuming knock down experiments typically necessary to understand the full impact of a TF would have a huge impact on our ability to parse out causative regulatory information. To speak colloquially for a moment, the ability to adjust one lever of the transcriptomic architecture and model how the rest will adjust is, simply put, cool.

That said, the actual functional implications of this technique are a bit more nebulous. The authors utilize these perturbations to predict gene essentiality scores as a method of identifying regulatory TFs that define GBM subtypes based on differential activity. However, the VIPER algorithm] already permits similar analyses and has been used to predict master regulators (MRs) that control cell state across a variety of malignancies. It is unclear whether inTRINISIC provides a functional improvement over existing methodology in terms of identifying these MRs, and the authors do not perform any comparisons to identify the differences between them.

Going further, the ability of this methodology to cut down on the complexity of large scale perturbational assays is not clear either. For instance, TFs are typically not directly targetable by drugs, and predicting which drugs will affect which TFs in a given context typically requires a perturbational assay in a sample as cell line or tissue as close to the sample of interest as possible. As such, while this method should be able to predict how a given sample's transcriptional architecture will change if a TF is knocked down, bridging the gap to actionable clinical information doesn't seem possible. To be clear, the author's have made no such claims, but this avenue was one of the immediate possible implications that came to mind upon reading the paper that unfortunately does not seem viable upon further scrutiny.

We thank the reviewer for their very thoughtful and encouraging comments. In response to their suggestions, we focused on two major issues : 1) Providing significantly greater detail and analysis regarding how this approach compares with and improves upon existing methodology and 2) Providing additional validation of putative biological cancer drug targets identified using this approach. Here, we used both DepMap perturbational data sets, as well as additional GBM clinical data. We completely concede that the identification of a TF dependency may be a long way from developing a drug that can target this dependency. However, TFs are increasingly the subject of drug development efforts, including the Myb-like TF identified in this study. This important issue is expanded upon in this discussion.

Major Points

In the first step of the inTRINISIC pipeline, previously existing regulatory networks are combined with both open chromatin data and known motif-binding maps. The first half of that formula - the pre-existing networks - come from brain tissues in the FANTOM5 project. While these networks are no doubt more relevant to the GBM samples discussed, there are still major transcriptomic differences between healthy and cancerous tissue. Meanwhile, the open chromatin data (from ENCODE) and the motif-binding information (from JASPAR) suffer from similar problems of

context specificity, though the cell lines in question are likely a better model for the GBM tissue being studied than healthy tissue.

We thank the reviewer for this comment, and we have clarified the rationale for our backbone network construction method in the revised manuscript. We used two distinct types of networks related to central nervous system cells, one from normal brain tissues and another inferred from open chromatin data in brain tumor cell lines. These networks were used to: 1) refine candidate edges to those most relevant to our tissue of interest (to reduce the dimensionality in our regression model) and 2) at the same time, ensure a more comprehensive coverage of the possible edges that our model can take into consideration - since tumor cells are also expected to retain at least part of the lineage programs inherited from their cell type of origin.

However, this approach based on pre-existing data for TFs with known binding motifs opens up an obvious hole in the methodology; no proteins with unknown binding motifs can be studied. The final transcriptomic network contains 518 TFs, but the human genome contains roughly 1800 TFs and an additional 600 co-TFs that all can play major roles as checkpoint modulators. Finally, the method of mapping from motif-binding graphs to a regulator-target graph - based on the nearest gene method often employed in ATACseq or other similar techniques - is prone to even more issues related to context specificity, as well as an inability to resolve more complicated regulatory structures in the chromatin itself].

All of these issues can be boiled down to one core issue; relying almost entirely on the literature for network construction is problematic due to the various bias associated with the construction of this prior knowledge. These biases are unavoidable with the method presented in the paper, but comparisons with more unbiased methods of network reconstruction are warranted. Various correlation based network reconstruction algorithms, machine learning based methods (SCENIC), or mutual information based (ARACNe)] are all publicly available and could be used to infer networks for comparison to the method proposed in the paper. It's difficult to go so far as to say which method - biased or unbiased - is superior in a vacuum, but the authors should at least acknowledge some of the limitations inherent in their approach and address them in some manner.

We appreciate the reviewer's insight here, and we address these issues more clearly in our revised manuscript. In our Supplementary Information, we have compared the performance of our models with the commonly used linear regression method for gene regulatory network inference. Here, we show a general superiority of our method in terms of recovering known regulatory interactions. In our revised manuscript we have provided a comparison with ARACNe (Figure 2 and Supplementary Figure 2) and discussed relative strengths and weaknesses of inTRINSiC compared to SCENIC (Discussion). It is noteworthy that networks built from ARACNe can no longer be applied to our downstream modeling of protein network activity, since it is not designed for predicting gene expression in response to perturbation.

As a final note, the author's critiques of existing methods for regulatory network generation are not entirely accurate. ARACNe, for instance, has a pair of parameters that capture both the

relative certainty of an edge's existence and the directionality of its regulation. Taken together - along with ARACNe's extremely low false positive rate - these two parameters essentially capture the relative strength of the TF-target interaction. Similar to the previous paragraph, flaws in the manuscript's critique of other network generation methods could be resolved via a more direct comparison between them and a deeper discussion of the pros and cons of this novel method.

We thank the reviewer for pointing this out. As mentioned above, we have included a more detailed comparison between our methodology and existing approaches to better clarify the pros and cons of our models.

Moving past the generation of the network framework, the use of non-linear regression to parametrize regulatory edges is interesting and appropriate. A brief comment discussing why a method that included cross terms for co-regulatory effects was either not possible or not attempted would be relevant here. The subsequent method of determining the correlation of regulators in order to determine co-regulatory proteins also seems robust in theory. However, the use of the RANSAC algorithm - a technique designed more to leave out outliers due to poor data collection or signal loss - seems inappropriate here, as the vectors being correlated are not raw data but an inferred statistic. If RANSAC was implemented because methods based purely on correlation were not sufficient, that could indicate a problem with overfitting in this context.

This is an important issue that we clarify in the revised manuscript. Cross terms for co-regulatory effects were an initial consideration but would be computationally prohibitive for our regression models, especially if we were to move beyond second-order interactions. However, our method did still retain part of that information, since we still see strong correlation structures in the resulting F value profiles.

As for the choice of RANSAC, despite it being commonly used for outlier detection, here we do not conceptually categorize data points filtered out by RANSAC as 'outliers' in the observations (in this case parameters, i.e. F values generated in our regression models). Rather, we use RANSAC to determine the local correlation structure between the regulatory activity of two transcription factors at a *subset* of target genes. The rationale behind this is that, even for broadly co-regulatory transcription factors, it is not unusual for them to act independently of each other in the modulating expression of some of their target genes. In addition, the use of RANSAC helps guard against highly skewed correlation computation arising from a small proportion of extreme values often present in the estimated F values.

Minor Points

Beyond the more significant challenges listed to the methodology discussed above, there are a few areas where explanations could be more clear, or where there has simply been an editing oversight. These are listed in bulleted form below:

The explanation in the Results section about subtype-specific regulatory profiles, particularly the explanation about regulatory signatures versus co-regulatory signatures, is a bit unclear. A reorganization that first explains the regulatory signature, then the co-regulatory signature, then the comparison between them - perhaps in different sections - might be easier to interpret.

We thank the reviewer for the constructive comment. We have made the recommended changes in the revised manuscript.

The reference for the PageRank algorithm is missing on page 11.

We apologize for the omission and have added the following references to the PageRank algorithm.

Alvarez MJ, Shen Y, Giorgi FM, et al. Functional characterization of somatic mutations in cancer using network-based inference of protein activity. *Nat Genet.* 2016;48(8):838-847. doi:10.1038/ng.3593

Aibar S, González-Blas CB, Moerman T, et al. SCENIC: single-cell regulatory network inference and clustering. *Nat Methods.* 2017;14(11):1083-1086. doi:10.1038/nmeth.4463

Lachmann A, Giorgi FM, Lopez G, Califano A. ARACNe-AP: gene network reverse engineering through adaptive partitioning inference of mutual information. *Bioinformatics.* 2016;32(14):2233-2235. doi:10.1093/bioinformatics/btw216

RE: Manuscript MSB-20-9506R, Integrated regulatory models for inference of subtype-specific susceptibilities in glioblastoma

Thank you again for sending us your revised manuscript. We have now heard back from the three reviewers who were asked to evaluate your revised study. As you will see below, the reviewers still raise substantial concerns on your work, which preclude its publication in *Molecular Systems Biology*.

While reviewer #1 is supportive, unfortunately reviewers #2 and #3 are not convinced that the performed revisions have adequately addressed several of the previously raised major issues. Specifically, they still raise concerns regarding the overall level of validation and think that it seems insufficient to raise confidence in the performance of the approach and its ability to generate biologically relevant predictions. Moreover, reviewer #3 points out that the superiority and advantages of inTRINSIC compared to alternative approaches are not convincingly demonstrated. As such, the reviewers indicated that they do not support publication of the study in *Molecular Systems Biology*.

Overall, and considering that our editorial policy allows in principle a single round of major revision, I am afraid I see no choice but to return the manuscript with the message that we cannot offer to publish it. I am sorry that the review of your work did not result in a more favorable outcome on this occasion, but I hope that you will not be discouraged from sending your work to *Molecular Systems Biology* in the future. In any case, thank you for the opportunity to examine this work.

Reviewer #1:

Liu et al have addressed my concerns and the robustness and timeliness of the study have been improved by adding new cross-validation data and computational methods. I feel that a further substantial revision may not be required; however, I strongly recommend additional editing in terms of improving data presentation and transparency of reporting.

Examples (main figures):

- Please provide a Y-axis for Figure 1D (correlation coefficients?) - the legend should include a statement as to how the p-values were calculated.
 - Please clarify whether Kaplan Meier plots show 'overall survival' (not recurrence-free survival)?
 - Is a font size increase possible for Figure 1 (e) and Figure 2 (e)?
 - Figure 4 legend (c): 'TF pairs with F value correlation coefficients larger than 0.8 are visualized as links': what is meant by 'visualized as links' (not clear to me)? Will hyperlinks be included?
 - Figure 5 legend: '(d) Bar plots of DepMap (grey) and predicted (color-coded according to subtypes as in (c))': panel (c) does not seem to include colors/subtypes?
 - Figure 5: axis labelling appears to be missing for (b), (f); difficult to read axis labels in (c), (d). Can consistent labelling be used across the panels?
 - Figure 5 legend: what does '* - p = 0.044' mean? Correct?
- Figure 5 (e): Please consider rephrasing following statement with regards to improving clarity: 'Shown are the sixTFs that has the same subtype that is most dependent on its expression (predicted by inTRINSiC) and that which shows the most negative correlation between expression and patient survival'.

Reviewer #2:

Manuscript Number: MSB-20-9506R

Manuscript Title: Integrated regulatory models for inference of subtype-specific susceptibilities in glioblastoma

It is clear that the authors have made progress in addressing some of the concerns raised in the original review. Particularly, the authors do make an effort to validate their model predictions by comparing their model predictions against independent data sets (CCGA and DepMap), focusing on a few TFs such as NFE2, MYBL2, NRF1, and ATF4 and comparing predicted vs. previously determined essentiality scores. A good example of the computational validation performed is presented in Figure 5g, showing the stratification of patient survival based on MYBL2, a TF identified from their network model. Unfortunately, there are still concerns associated with validation that dampen enthusiasm for the revised manuscript:

Major concerns/questions:

- The authors claim to use DepMap data as a means to validate predictions of TF and protein level activity, however it seems that the DepMap essentiality scores are used to fit protein activity scores. If the DepMap scores are used to fit model outputs to decrease overfitting and ensure

sparsity, as described in the "Modeling effects of gene expression perturbation in silico", then this is not really a true computational validation of the model predictions. Further, there need to be a number of tests that need to be run, including the shuffling of node labels on the iNTRINSiC regulatory network and determining the threshold for a significant score, a hypergeometric test, etc. The way they make their essentiality prediction, you recursively propagate the effect of a TF knockout through the network of TF-TF interactions. So, if you hit a hub TF then any TF could be deemed essential. So a network where node labels are shuffled, and TF and gene in/outdegree is maintained will essentially yield a similar number of essential TFs (at least I would expect it to). The distribution of essential/non-essential TFs could therefore look similar. In that regard a ROC analysis with leave-one-out permutations would also be necessary to show sensitivity and specificity (and associated significance). If the model predictions of TF essentiality were calculated independently and then compared to DepMap, that would be better. If that was indeed the case, then that needs to be clarified further. As it is currently written, it is not clear that is the case.

- I'm concerned with the statement made about the relatively low impact that miRNAs have on expression regulation:

"1) the effects of miRNA regulation are estimated to explain only 7-13% of overall gene expression variation⁴⁹, and 2) since miRNAs act primarily via repressing mRNA levels⁵⁰, regulation by miRNAs may be absorbed into TF regulation in our model where a subset of TF-target regulatory parameters may be due to indirect regulation of the target by TFs through expression of miRNAs⁵⁰."

While this may be true regarding the relationship between miRNAs and overall gene expression variation, this may not hold for specific genes and miRNAs. Context plays a role and to brush a broad stroke on the perceived minimal impact of miRNAs on expression regulation may lead to incorrect conclusions of a particular TFs role in gene regulation. It seems that the role of miRNA regulation is being confounded in the model. While one cannot account for every single detail of regulation in a quantitative model, as it would be intractable, this raises a concern.

- Figure 1d - the effort to show the similarities between bulk- and single-cell expression data via correlation of F values is appreciated. However, there still seems to be a good portion of the correlation relationships that are either non-correlated or anti-correlated. This is not the most convincing figure to show the similarity between bulk and single-cell data. Is there some permutation analysis or statistical analysis to show that the number of F-correlations is indeed statistically significant? If so, at what correlation level are the F-correlations between bulk and single-cell data statistically significant? It would be interesting to see what TF-TF relationships are anti-correlated or have no relationship and if that is indicative of some technical or biological factor.

- The patient stratification of low and high MYBL2 looks good in the Kaplan-Meier plots (Fig 5G), but the significance seems somewhat low (p-value (TCGA): 0.035 and p-value (CGGA): 0.0228).

- Ultimately, to be convinced, I would need some targeted experiments to validate new subtype-specific essential TFs.

Minor comments:

- There are quite a few separate algorithms used as part of the overall workflow to generate a regulatory network model and predictions. It would be helpful if some type of flow-diagram were included, one that is a bit more detailed than the one included in Fig 1a. A supplementary figure that

were to include in some way, shape, or form a general overview of the various algorithms, databases, and steps involved (e.g., PIQ algorithm, RANSAC algorithm, BIOGRID database, etc., etc., etc.).

- Supplementary Fig 1F is missing - no histograms showing overlap between TF-edges derived from TCGA and CGCA data are shown.
- Supplementary Fig 4C, which shows the percentile ranking of NFE2 essentiality across CL, PN, and MES subtypes, is not convincing. I would want to see a targeted essentiality test on NEF2 knockdown in different GBM subtype cells.
- Figure 5B axes are not labeled
- Supplementary Figure 1e legend - is "Mean symmetric mean absolute percentage error" correct? "Mean" is used twice in the sentence and may be a typo.

Reviewer #3:

In this revised manuscript, the authors have adequately addressed several points raised in our earlier review. However, some key concerns remain unaddressed, which will be discussed in the following. First, let's start by acknowledging the points that have been addressed:

First, the authors now adequately explain why it was not viable to include cross-terms in their model. Also, while the use of RANSAC was questioned in the original review, the rebuttal clarifies how it was applied and shows that it is appropriate for their data. Finally, even though this was not specifically requested, in the original review, the new Figure S2e does a great job of illustrating how the in silico perturbational assay can move cells in gene expression space.

This reviewer had also questioned the lack of comparative analysis of inTRINSiC to other established methods, such as ARACNe and SCENIC. As a result, it was difficult to weigh both benefits and drawbacks associated with the new algorithm and, in particular, to accurately assess whether it provided any improvements relative to the current state of the art. To address these concerns, the authors now utilize the BIOGRID database to explicitly compare the performance of ARACNe and inTRINSiC, concluding that they have comparable performances. They also clarify what they consider to be the specific advantages of their method. In particular, they illustrate inTRINSiC's ability to model the effect of downstream effectors on gene expression.

While these represents key improvements on the original manuscript submission, some key concerns remain unaddressed. For instance, the comparison between ARACNe and inTRINSiC relies largely on TF-TF interactions as a gold standard baseline, thus ignoring the vast majority of interactions which are between TFs and their transcriptional targets. BioGRID provides additional data for interactions between TFs and known targets, which could have been used for this purpose. There are also additional databases, such as TRRUST, which report information about TF-target interactions. Given the general availability of these data, why was the comparison performed using only a relatively restricted set?

While the authors discuss what they believe to be the advantages of the inTRINSiC approach relative to SCENIC in the discussion section, it appears that no direct comparison of their relative

performance was made. As a result, for two of the mainstream approaches, it appears that one is essentially equivalent to the new methodology while the other was not tested at all.

More importantly, the authors have not explored any further validation of their downstream predictions, either experimentally or against other methods that identify key TFs that control cell state. This becomes more critical given that the ability of inTRINSiC to outperform other methods in terms of network reconstruction is not obvious from the new data. For instance, an obvious comparison would be to the SCENIC and VIPER algorithms which could also be benchmarked against DepMap, as done in Figure 5. This would establish an objective baseline for differential performance between inTRINSiC predictions and predictions generated by other methods. This is particularly critical because some of the predictions by inTRINSiC are not consistent with experimental validation. For instance, the algorithm predicts that silencing STAT3 in mesenchymal glioma samples would shift the phenotype towards a proneural phenotype. However, in the manuscript they cite (Carro et al. Nature 2010), experimental silencing of STAT3 alone failed to reprogram mesenchymal cells towards a proneural state and, as computationally predicted, joint silencing of both STAT3 and CEBPB was required to accomplish this goal.

Similarly, while the example of MYBL2 is discussed at length, the authors do not provide any experimental validation to support these findings. Indeed, all validation attempts provided by the manuscript are completely retrospective, e.g. using DepMap data. As a result, contrary to what has been done by numerous manuscripts that have used computational approaches to prioritize cell dependencies or reprogramming factors and then validated these findings experimentally, this manuscript fails to provide any level of experimental validation supporting the claim that inTRINSiC predictions can discover new biology.

Finally, I still feel that the authors do not adequately describe the real implications and value of inTRINSiC. The ability to perturb gene expression profiles in silico is novel and potentially interesting, but the authors present the algorithm as a way to prioritize essential TFs governing biological phenotypes and that would require either experimental validation to substantiate their claims or comparison to experimentally validated algorithms that perform similar tasks. As a result, proposing that inTRINSiC may be able to predict treatments entirely in silico seems tenuous and unsubstantiated by the data, especially given the difficulty of directly targeting TFs, a point the authors concede in the rebuttal, but do not discuss at all in the revised manuscript.

Thank you for the opportunity to respond to the reviewers' concerns. As I mentioned in our previous correspondence, we believe that much of the final opinion of reviewers #2 and #3 was motivated by the lack of requested cell culture-based experimental validation. Typically, if reviewers suggest experiments, we are anxious to perform them, and I would be highly critical of a revised manuscript that failed to address an experimental issue that I raised in review. This is particularly confounding, as my group has a major focus on in vivo experimental models of cancer therapy – work that has ground to a halt for the last 5 months. That being said, we believe that the DepMap data represents an extensive experimental data set that we have been able to use as a key validation tool.

For the remainder of the reviewer concerns, we think there was some misunderstanding regarding The DepMap validation work, the inTRINSiC algorithm and data that is already present in this study. We have included a point by point response to all reviewer concerns. Again, we thank you for your editorial supervision of this work. We are more than ever convinced of the value of this approach to the cancer community and would be very excited to have this work published in MSB.

Reviewer #1:

Liu et al have addressed my concerns and the robustness and timeliness of the study have been improved by adding new cross-validation data and computational methods. I feel that a further substantial revision may not be required; however, I strongly recommend additional editing in terms of improving data presentation and transparency of reporting.

Examples (main figures):

- Please provide a Y-axis for Figure 1D (correlation coefficients?) - the legend should include a statement as to how the p-values were calculated.
 - Please clarify whether Kaplan Meier plots show 'overall survival' (not recurrence-free survival)?
 - Is a font size increase possible for Figure 1 (e) and Figure 2 (e)?
 - Figure 4 legend (c): 'TF pairs with F value correlation coefficients larger than 0.8 are visualized as links': what is meant by 'visualized as links' (not clear to me)? Will hyperlinks be included?
 - Figure 5 legend: '(d) Bar plots of DepMap (grey) and predicted (color-coded according to subtypes as in (c)': panel (c) does not seem to include colors/subtypes?
 - Figure 5: axis labelling appears to be missing for (b), (f); difficult to read axis labels in (c), (d). Can consistent labelling be used across the panels?
 - Figure 5 legend: what does '* - p = 0.044' mean? Correct?
- Figure 5 (e): Please consider rephrasing following statement with regards to improving clarity: 'Shown are the sixTFs that has the same subtype that is most dependent on its expression (predicted by inTRINSiC) and that which shows the most negative correlation between expression and patient survival'.

We agree with the reviewer's comments about the "robustness and timeliness" of this study. All of the data presentation changes can be made.

Reviewer #2:

Manuscript Number: MSB-20-9506R

Manuscript Title: Integrated regulatory models for inference of subtype-specific susceptibilities in glioblastoma

It is clear that the authors have made progress in addressing some of the concerns raised in the original review. Particularly, the authors do make an effort to validate their model predictions by comparing their model predictions against independent data sets (CCGA and DepMap), focusing on a few TFs such as NFE2, MYBL2, NRF1, and ATF4 and comparing predicted vs. previously determined essentiality scores. A good example of the computational validation performed is presented in Figure 5g, showing the stratification of patient survival based on MYBL2, a TF identified from their network model. Unfortunately, there are still concerns associated with validation that dampen enthusiasm for the revised manuscript:

Major concerns/questions:

- The authors claim to use DepMap data as a means to validate predictions of TF and protein level activity, however it seems that the DepMap essentiality scores are used to fit protein activity scores. If the DepMap scores are used to fit model outputs to decrease overfitting and ensure sparsity, as described in the "Modeling effects of gene expression perturbation in silico", then this is not really a true computational validation of the model predictions. Further, there need to be a number of tests that need to be run, including the shuffling of node labels on the inTRINSiC regulatory network and determining the threshold for a significant score, a hypergeometric test, etc. The way they make their essentiality prediction, you recursively propagate the effect of a TF knockout through the network of TF-TF interactions. So, if you hit a hub TF then any TF could be deemed essential. So a network where node labels are shuffled, and TF and gene in/outdegree is maintained will essentially yield a similar number of essential TFs (at least I would expect it to). The distribution of essential/non-essential TFs could therefore look similar. In that regard a ROC analysis with leave-one-out permutations would also be necessary to show sensitivity and specificity (and associated significance). If the model predictions of TF essentiality were calculated independently and then compared to DepMap, that would be better. If that was indeed the case, then that needs to be clarified further. As it is currently written, it is not clear that is the case.

There are two issues here. First, how did we perform our DepMap validation - and is it a true independent validation? Second, by looking at hub TFs, won't all transcription factors be deemed essential?

For the first issue, we would like to clarify that the design of the essentiality prediction part of inTRINSiC is to test if a particular combination of transcription regulatory and protein signaling activities could be used to predict the changes in fitness of the cell upon perturbing each TF. This involves, for any data set or tumor type, developing parameters for the essentiality prediction portion. The DepMap represents a powerful dataset for mining biological pathways that best predict TF perturbation outcomes, but it still needs to be parameterized for use. Thus, we generated training data using some data from the DepMap and then validated our findings using a completely independent set of data from the DepMap. It is, in fact, independent validation, as is performed in many contexts on many data sets. To further clarify our point, we would also like to emphasize that for each particular type of tissue/tumor, the inTRINSiC pipeline will have to be run independently to obtain specific parameters for the essentiality prediction portion. We could put in a supplementary figure to graphically represent our work flow.

For the second issue, we agree that perturbing a TF that acts through a hub TF would potentially deem that TF significant. However, we would like to highlight two observations from our data that already address this concern: 1) perturbing many of the known hub/critical TFs themselves did not result in a striking amount of fitness decrease, for example MYC, which has thousands of regulatory targets and 2) Hub TFs, in fact, show different levels of essentiality in different tumor subtypes. Thus, we are able to capture differential biology between subtypes even in the case of hub TFs.

• I'm concerned with the statement made about the relatively low impact that miRNAs have on expression regulation:

"1) the effects of miRNA regulation are estimated to explain only 7-13% of overall gene expression variation⁴⁹, and 2) since miRNAs act primarily via repressing mRNA levels⁵⁰, regulation by miRNAs may be absorbed into TF regulation in our model where a subset of TF-target regulatory parameters may be due to indirect regulation of the target by TFs through expression of miRNAs⁵⁰."

While this may be true regarding the relationship between miRNAs and overall gene expression variation, this may not hold for specific genes and miRNAs. Context plays a role and to brush a broad stroke on the perceived minimal impact of miRNAs on expression regulation may lead to incorrect conclusions of a particular TFs role in gene regulation. It seems that the role of miRNA regulation is being confounded in the model. While one cannot account for every single detail of regulation in a quantitative model, as it would be intractable, this raises a concern.

We agree with the reviewer that inclusion of a systematic map of miRNA-TF-gene interactions would be intractable for the scope of this paper. Indeed, this analysis would require a different type of quantitative modeling. However, we would emphasize that our approach is highly validated in the DepMap data set in the absence of miRNA data. While inclusion of miRNA may refine our data, we do not believe it will fundamentally alter our conclusions.

• Figure 1d - the effort to show the similarities between bulk- and single-cell expression data via correlation of F values is appreciated. However, there still seems to be a good portion of the correlation relationships that are either non-correlated or anti-correlated. This is not the most convincing figure to show the similarity between bulk and single-cell data. Is there some permutation analysis or statistical analysis to show that the number of F-correlations is indeed statistically significant? If so, at what correlation level are the F-correlations between bulk and single-cell data statistically significant? It would be interesting to see what TF-TF relationships are anti-correlated or have no relationship and if that is indicative of some technical or biological factor.

We have calculated p -values for the correlation coefficient distributions using a Mann-Whitney U test to test if the medians deviate significantly from zero, and they are significant for all 3 subtypes. We would be sure to inspect the correlations that are near-zero or negative in a revised manuscript.

• The patient stratification of low and high MYBL2 looks good in the Kaplan-Meier plots (Fig 5G), but the significance seems somewhat low (p -value (TCGA): 0.035 and p -value (CGGA): 0.0228.

We would like to point out that the diminished significance is likely due to the relatively small sample sizes of these cohorts due to subsetting to the Proneural subtype. We are however confident that consistent observations in two completely independent datasets of two profiling platforms (micro-array and RNA-seq) and different demographics (largely non-Asian American and Chinese) are sufficient to prioritize MYBL2 for further investigation. Moreover, the p -value of <0.05 is a well-established cutoff for statistical significance.

- Ultimately, to be convinced, I would need some targeted experiments to validate new subtype-specific essential TFs.

The reviewer clearly expects experimental validation, which we have been unable to perform.

Minor comments:

- There are quite a few separate algorithms used as part of the overall workflow to generate a regulatory network model and predictions. It would be helpful if some type of flow-diagram were included, one that is a bit more detailed than the one included in Fig 1a. A supplementary figure that were to include in some way, shape, or form a general overview of the various algorithms, databases, and steps involved (e.g., PIQ algorithm, RANSAC algorithm, BIOGRID database, etc., etc.).
- Supplementary Fig 1F is missing - no histograms showing overlap between TF-edges derived from TCGA and CGCA data are shown.
- Supplementary Fig 4C, which shows the percentile ranking of NFE2 essentiality across CL, PN, and MES subtypes, is not convincing. I would want to see a targeted essentiality test on NEF2 knockdown in different GBM subtype cells.
- Figure 5B axes are not labeled
- Supplementary Figure 1e legend - is "Mean symmetric mean absolute percentage error" correct? "Mean" is used twice in the sentence and may be a typo.

All of these changes can be made and information provided, with exception of NEF2 knockdown experiments.

Reviewer #3:

In this revised manuscript, the authors have adequately addressed several points raised in our earlier review. However, some key concerns remain unaddressed, which will be discussed in the following. First, let's start by acknowledging the points that have been addressed:

First, the authors now adequately explain why it was not viable to include cross-terms in their model. Also, while the use of RANSAC was questioned in the original review, the rebuttal clarifies how it was applied and shows that it is appropriate for their data. Finally, even though this was not specifically requested, in the original review, the new Figure S2e does a great job of illustrating how the in silico perturbational assay can move cells in gene expression space.

We appreciate the reviewer's enthusiasm here.

This reviewer had also questioned the lack of comparative analysis of inTRINSiC to other established methods, such as ARACNe and SCENIC. As a result, it was difficult to weigh both benefits and drawbacks associated with the new algorithm and, in particular, to accurately assess whether it provided any improvements relative to the current state of the art. To address these concerns, the authors now utilize the BIOGRID database to explicitly compare the performance of ARACNe and inTRINSiC, concluding that they have comparable performances. They also clarify what they consider to be the specific advantages of their method. In particular, they illustrate

inTRINSiC's ability to model the effect of downstream effectors on gene expression.

While these represents key improvements on the original manuscript submission, some key concerns remain unaddressed. For instance, the comparison between ARACNe and inTRINSiC relies largely on TF-TF interactions as a gold standard baseline, thus ignoring the vast majority of interactions which are between TFs and their transcriptional targets. BioGRID provides additional data for interactions between TFs and known targets, which could have been used for this purpose. There are also additional databases, such as TRRUST, which report information about TF-target interactions. Given the general availability of these data, why was the comparison performed using only a relatively restricted set?

We chose TF-TF interaction as a standard for evaluating model performance as it contains a large body of experimental evidence. In contrast, the BioGRID data set is derived largely from predicted TF binding sites and physical binding experiments. Thus, our analysis focuses on a more extensively validated subset of the overall BioGRID data set. We would be happy to provide a parallel analysis using BioGRID regulatory interactions as reference standards to benchmark our networks, although we are quite confident in the superiority of our data set.

While the authors discuss what they believe to be the advantages of the inTRINSiC approach relative to SCENIC in the discussion section, it appears that no direct comparison of their relative performance was made. As a result, for two of the mainstream approaches, it appears that one is essentially equivalent to the new methodology while the other was not tested at all.

This data is already included in our manuscript. When comparing our models with other methods, we have taken into consideration the major categories of regulatory network construction methods. SCENIC essentially runs a correlation-based pipeline to unveil regulatory structures from single-cell expression profiles. We have provided in our supplementary materials (Figure S1c and Supplementary Methods) a comparison between our method and a purely linear correlation-based method, which captures the same type of variation as SCENIC does.

More importantly, the authors have not explored any further validation of their downstream predictions, either experimentally or against other methods that identify key TFs that control cell state. This becomes more critical given that the ability of inTRINSiC to outperform other methods in terms of network reconstruction is not obvious from the new data. For instance, an obvious comparison would be to the SCENIC and VIPER algorithms which could also be benchmarked against DepMap, as done in Figure 5. This would establish an objective baseline for differential performance between inTRINSiC predictions and predictions generated by other methods. This is particularly critical because some of the predictions by inTRINSiC are not consistent with experimental validation. For instance, the algorithm predicts that silencing STAT3 in mesenchymal glioma samples would shift the phenotype towards a proneural phenotype. However, in the manuscript they cite (Carro et al. Nature 2010), experimental silencing of STAT3 alone failed to reprogram mesenchymal cells towards a proneural state and, as computationally predicted, joint silencing of both STAT3 and CEBPB was required to accomplish this goal.

We would like to emphasize that, as stated in our Discussion, other network inference algorithms, including SCENIC, does not allow for predictive gene expression modeling, i.e. given a perturbation, predict the changes in the expression levels of downstream regulatory targets. A comparison with the VIPER algorithm might be useful for evaluating the performance of the protein network activity prediction method in our paper. This could be provided.

We believe the reviewer has misread our data here. For the biology associated with STAT3, we would like to clarify that in our computational experiments, perturbing STAT3 caused a

substantial shift towards the Classical subtype (Figure 3e) rather than the Proneural subtype (as argued by the reviewer). We also observed that some Proneural subtypes have also shifted towards the Mesenchymal subtype in Figure 3e upon loss of STAT3. We argue that this is consistent with two observations: 1) STAT3 has significant participation in both Mesenchymal and Proneural signature sets (Figure 3b), and 2) STAT3 is a part of the Mesenchymal master regulatory circuit and is required for maintaining the Mesenchymal state, which is shown in Carro et al (Nature 2010), Figure 4 where knocking down of either STAT3 or CEBPB would lead to a loss of Mesenchymal signature gene expression.

Similarly, while the example of MYBL2 is discussed at length, the authors do not provide any experimental validation to support these findings. Indeed, all validation attempts provided by the manuscript are completely retrospective, e.g. using DepMap data. As a result, contrary to what has been done by numerous manuscripts that have used computational approaches to prioritize cell dependencies or reprogramming factors and then validated these findings experimentally, this manuscript fails to provide any level of experimental validation supporting the claim that inTRINSiC predictions can discover new biology.

Again, the reviewer clearly expects experimental validation, which we have been unable to perform.

Finally, I still feel that the authors do not adequately describe the real implications and value of inTRINSiC. The ability to perturb gene expression profiles in silico is novel and potentially interesting, but the authors present the algorithm as a way to prioritize essential TFs governing biological phenotypes and that would require either experimental validation to substantiate their claims or comparison to experimentally validated algorithms that perform similar tasks. As a result, proposing that inTRINSiC may be able to predict treatments entirely in silico seems tenuous and unsubstantiated by the data, especially given the difficulty of directly targeting TFs, a point the authors concede in the rebuttal, but do not discuss at all in the revised manuscript.

We have perhaps understated the relevance and utility of inTRINSiC in the manuscript. The framework for predictive modeling of gene essentiality provided by inTRINSiC will not only serve as a powerful platform for predicting treatment outcome in silico, but will also prove highly valuable in at least two areas: First, it will allow one to leverage large-scale profiling datasets from primary human tumor samples where unbiased genetic screens are infeasible to perform. This analysis will identify “high value” genes and pathways to target in cell line or mouse/xenograft models. Second, with improvements in the scale and coverage of newer techniques such as Perturb-seq and single-cell epigenomics, the inTRINSiC pipeline can be extended to understand the source of phenotypic heterogeneity at the single-cell level through modeling multiple layers of regulation in the same cell.

Finally, while targeting TFs themselves is challenging, such efforts are increasingly common and, as in the case of Brd4 inhibitors, represent a clinical reality.

Manuscript Number: MSB-20-9506RR-Q, Integrated regulatory models for inference of subtype-specific susceptibilities in glioblastoma

Thank you for your message asking us to reconsider our decision on your manuscript MSB-20-9506. I have now had the chance to evaluate the points raised in your appeal letter and I have also discussed them with the editorial team. As I will explain below, we have decided to give you the chance to revise the study, in an exceptional final round of revision.

The main reasons underlying our previous decision to decline publication were the issues raised by the reviewers on the overall level of validation and the somewhat limited evidence of superiority/advantages of iNTRINSiC compared to alternative approaches. I would like to stress out that our decision to decline publication was not based on the lack of follow up experimental validations, as we had discussed before the submission of the revised version that performing such validations would be exceptionally difficult due to the Sars-CoV-2 situation. However, in addition to reiterating their disappointment about the lack of follow up experimentation, in this round of review the reviewers also raised concerns on the computational aspects of the study which we thought represented significant limitations and dampened the confidence in the performance of the approach and its ability to generate biologically relevant predictions.

After having read your preliminary point by point response to the reviewers' concerns and given the overall supportive comments of the reviewers about the potential value of the approach (during both review rounds), we have decided to give you the chance to revise the study and to address the remaining issues raised by the referees regarding the performance of the method and its advantages over previous approaches. Specifically, it would be important to address the following:

- The superiority and advantages of iNTRINSiC compared to SCENIC and ARACNE need to be better explained and supported. Any features of iNTRINSiC that allow analyses that are not possible to perform with alternative approaches should be clearly described so that it is clear to the reader how this method goes beyond what is currently available. Any further computational analyses that could provide support for the performance of iNTRINSiC (e.g. comparison to VIPER) would be beneficial.
- The answer regarding the use of DepMap data for validation seems satisfactory. We would ask you to make sure that it is clearly described in the manuscript that independent data was used for training and validation.
- An additional Figure depicting the iNTRINSiC workflow in detail would be very beneficial for the readers.
- Regarding the issue related to the effect of miRNAs, we think that no further analyses are required. It is sufficient to include a short discussion on the expected minor effect of miRNAs on the conclusions.

- As we had previously discussed we think that follow up experimental validations are not mandatory for the acceptance of the work.

- All minor issues raised by the reviewers regarding the data presentation etc. need to be addressed.

Thank you again for the opportunity to submit a revised manuscript. We have included a detailed point-by-point response to your comments as well as those provided by the reviewers. We believe that we have been able to completely address all of the issues raised. As mentioned in our previous correspondence, there were a number of reviewer concerns that perhaps arose out of a misunderstanding of our approach or a lack of clarity on our part. We have endeavored to elaborate on these issues of confusion and provide graphical and table data presentation to provide additional clarity. We have also substantially extended our direct comparisons between iNTRINSiC and existing algorithms – analyses that highlight the clear advantages of our approach.

We are more than ever convinced of the value of this approach to the cancer community and are very excited to have this work published in MSB. We also truly appreciate your editorial supervision of this work. This manuscript has been greatly improved over the course of this process, and your guidance has significantly contributed to this process.

Editor's comments:

Dear Mike,

Thank you for your message asking us to reconsider our decision on your manuscript MSB-20-9506. I have now had the chance to evaluate the points raised in your appeal letter and I have also discussed them with the editorial team. As I will explain below, we have decided to give you the chance to revise the study, in an exceptional final round of revision.

After having read your preliminary point by point response to the reviewers' concerns and given the overall supportive comments of the reviewers about the potential value of the approach (during both review rounds), we have decided to give you the chance to revise the study and to address the remaining issues raised by the referees regarding the performance of the method and its advantages over previous approaches. Specifically, it would be important to address the following:

- The superiority and advantages of iNTRINSiC compared to SCENIC and ARACNE need to be better explained and supported. Any features of iNTRINSiC that allow analyses that are not possible to perform with alternative approaches should be clearly described so that it is clear to the reader how this method goes beyond what is currently available. Any further computational analyses that could provide support for the performance of iNTRINSiC (e.g. comparison to VIPER) would be beneficial.

In the revised manuscript, we now provide additional text and a table that directly compares iNTRINSiC with ARACNe, SCENIC and VIPER and describes the advantages of iNTRINSiC relative to these other approaches. We also provide additional computational comparisons

between 1) inTRINSiC and GENIE3 (another state-of-the-art network inference method used by the SCENIC single-cell pipeline), showing that they are comparable in terms of the number of BIOGRID interactions captured from the pipelines (Figure EV2d and Table EV1, see below), and 2) inTRINSiC and VIPER for protein signaling activity inference, *showing that inTRINSiC's exponential ranking algorithm achieves wider coverage of signaling proteins* and consistency with known (non-transcriptional regulatory) signaling relationships compared to VIPER (Figure EV2h-j).

Network Construction Method	inTRINSiC	ARACNe	SCENIC (GENIE3)	SYGNAL
Type of regulator-target relationship captured	Non-linear	Mutual information	Co-expression*	Co-expression
Pre-select regulator-target edges based on experimental evidence	Yes	No	No	Yes
Yields regulatory parameters with direct biological interpretations (magnitude and directionality etc.)?	Yes	No	No	No
Allows in silico perturbation of transcriptional regulators?	Yes	No	No	No
Allows explicit incorporation of additional mechanisms of transcription regulation (non-coding RNA, epigenetic etc.)?	Yes	No	No	Yes

Protein Activity Inference Method	inTRINSiC	VIPER
Utilizes protein-protein signaling interactions	Yes	No
Independent of transcription regulation inference?	Yes	No**

*: co-expression here is defined as a (linear or non-linear) relationship between the regulator and target genes inferred using random forest regression, where the relationship does not take any particular analytical form.

** : In an extension of the VIPER algorithm, residual post-translational signaling activities are inferred by regressing out transcriptional variance. However, the VIPER algorithm

still relies on differential expression to infer either direct or indirect regulatory effects on protein activity.

- The answer regarding the use of DepMap data for validation seems satisfactory. We would ask you to make sure that it is clearly described in the manuscript that independent data was used for training and validation.

We have changed the text to make it clear that we are using a completely different test cell line set than was used in our initial training. We also provide a figure that clearly depicts our DepMap validation workflow.

- An additional Figure depicting the inTRINSiC workflow in detail would be very beneficial for the readers.

We completely agree and now provide additional figures (5a and Figure EV5) that describe the inTRINSiC workflow.

- Regarding the issue related to the effect of miRNAs, we think that no further analyses are required. It is sufficient to include a short discussion on the expected minor effect of miRNAs on the conclusions.

We have added text to the discussion to explain the possible impact of miRNAs and our reasons for not including miRNA analysis in this study.

- As we had previously discussed we think that follow up experimental validations are not mandatory for the acceptance of the work.

We thank you for this consideration during this difficult time for our laboratories. While experimental validation, particularly *in vivo* validation in mouse/PDX models would be nice, we think our DepMap analysis is both decisive and of considerable benefit for future analyses. The reason for this is two-fold. First, it represents a large-scale wet bench/experimental data set that we have used to validate our approach. Second, it provides a work-flow by which any patient tumor expression set can be analyzed for subtype dependencies by combining inTRINSiC with DepMap. This is one of the things that makes our approach different from any other – the ability to take patient tumor samples (GBM or otherwise) that cannot be genetically modified or chemically screened and test for dependencies.

- All minor issues raised by the reviewers regarding the data presentation etc. need to be addressed.

All of the minor issues raised by the reviewers have been addressed (see below).

Reviewer #1:

Liu et al have addressed my concerns and the robustness and timeliness of the study have been improved by adding new cross-validation data and computational methods. I feel that a further substantial revision may not be required; however, I strongly recommend additional editing in terms of improving data presentation and transparency of reporting.

Examples (main figures):

- Please provide a Y-axis for Figure 1D (correlation coefficients?) - the legend should include a statement as to how the p-values were calculated.

-Please clarify whether Kaplan Meier plots show 'overall survival' (not recurrence-free survival)?

-Is a font size increase possible for Figure 1 (e) and Figure 2 (e)?
- Figure 4 legend (c): 'TF pairs with F value correlation coefficients larger than 0.8 are visualized as links': what is meant by 'visualized as links' (not clear to me)? Will hyperlinks be included?
-Figure 5 legend: '(d) Bar plots of DepMap (grey) and predicted (color-coded according to subtypes as in (c)': panel (c) does not seem to include colors/subtypes?
- Figure 5: axis labelling appears to be missing for (b), (f); difficult to read axis labels in (c), (d). Can consistent labelling be used across the panels?
-Figure 5 legend: what does '* - p = 0.044' mean? Correct?
Figure 5 (e): Please consider rephrasing following statement with regards to improving clarity:
'Shown are the six TFs that has the same subtype that is most dependent on its expression (predicted by iNTRINSiC) and that which shows the most negative correlation between expression and patient survival'.

We appreciate the reviewer's comments about the "robustness and timeliness" of this study. We have made numerous changes to improve the data presentation in the revised manuscript, including all of these suggested changes.

Reviewer #2:

Manuscript Number: MSB-20-9506R

Manuscript Title: Integrated regulatory models for inference of subtype-specific susceptibilities in glioblastoma

It is clear that the authors have made progress in addressing some of the concerns raised in the original review. Particularly, the authors do make an effort to validate their model predictions by comparing their model predictions against independent data sets (CCGA and DepMap), focusing on a few TFs such as NFE2, MYBL2, NRF1, and ATF4 and comparing predicted vs. previously determined essentiality scores. A good example of the computational validation performed is presented in Figure 5g, showing the stratification of patient survival based on MYBL2, a TF identified from their network model. Unfortunately, there are still concerns associated with validation that dampen enthusiasm for the revised manuscript:

Major concerns/questions:

- The authors claim to use DepMap data as a means to validate predictions of TF and protein level activity, however it seems that the DepMap essentiality scores are used to fit protein activity scores. If the DepMap scores are used to fit model outputs to decrease overfitting and ensure sparsity, as described in the "Modeling effects of gene expression perturbation in silico", then this is not really a true computational validation of the model predictions. Further, there need to be a number of tests that need to be run, including the shuffling of node labels on the iNTRINSiC regulatory network and determining the threshold for a significant score, a hypergeometric test, etc. The way they make their essentiality prediction, you recursively propagate the effect of a TF knockout through the network of TF-TF interactions. So, if you hit a hub TF then any TF could be deemed essential. So a network where node labels are shuffled, and TF and gene in/outdegree is maintained will essentially yield a similar number of essential TFs (at least I would expect it to). The distribution of essential/non-essential TFs could therefore look similar. In that regard a ROC analysis with leave-one-out permutations would also be necessary to show sensitivity and specificity (and associated significance). If the model predictions of TF essentiality were calculated independently and then compared to DepMap, that would be better. If that was indeed the case, then that needs to be clarified further. As it is currently written, it is not clear that is the case.

We think there are two separate and important issues here. First, how did we perform our DepMap validation - and is it a true independent validation? Second, by looking at hub TFs, won't all transcription factors be deemed essential?

For the first issue, we think there was a lack of clarity in our description. We would like to clarify that the design of the essentiality prediction part of inTRINSiC is to test if a particular combination of transcription regulatory and protein signaling activities could be used to predict the changes in fitness of the cell upon perturbing each TF. This involves, for any data set or tumor type, developing parameters for the essentiality prediction portion. The DepMap represents a powerful dataset for mining biological pathways that best predict TF perturbation outcomes, but it still needs to be parameterized for use. Thus, we generated training data using some data from the DepMap and then tested our findings using a completely independent set of data from the DepMap. It is independent validation, as is performed in many contexts on many data sets. To further clarify our point, we would also like to emphasize that for each particular type of tissue/tumor, the inTRINSiC pipeline will have to be run independently to obtain specific parameters for the essentiality prediction portion. We have modified the text for clarity and put in a supplementary figure to graphically represent our work flow.

For the second issue, we agree that perturbing a TF that acts through a hub TF would potentially deem that TF significant. However, we would like to highlight two observations from our data that already address this concern: 1) perturbing many of the known hub/critical TFs themselves did not result in a striking amount of fitness decrease, for example MYC, which has thousands of regulatory targets and 2) Hub TFs, in fact, show different levels of essentiality in different tumor subtypes. Thus, we are able to capture differential biology between subtypes even in the case of hub TFs.

• I'm concerned with the statement made about the relatively low impact that miRNAs have on expression regulation:

"1) the effects of miRNA regulation are estimated to explain only 7-13% of overall gene expression variation⁴⁹, and 2) since miRNAs act primarily via repressing mRNA levels⁵⁰, regulation by miRNAs may be absorbed into TF regulation in our model where a subset of TF-target regulatory parameters may be due to indirect regulation of the target by TFs through expression of miRNAs⁵⁰."

While this may be true regarding the relationship between miRNAs and overall gene expression variation, this may not hold for specific genes and miRNAs. Context plays a role and to brush a broad stroke on the perceived minimal impact of miRNAs on expression regulation may lead to incorrect conclusions of a particular TFs role in gene regulation. It seems that the role of miRNA regulation is being confounded in the model. While one cannot account for every single detail of regulation in a quantitative model, as it would be intractable, this raises a concern.

We agree with the reviewer that inclusion of a systematic map of miRNA-TF-gene interactions would be intractable for the scope of this paper. Indeed, this analysis would require a different type of quantitative modeling. However, we would emphasize that our approach is highly validated in the DepMap data set in the absence of miRNA data. We have added to the text on miRNAs in the discussion section to clarify these issues.

• Figure 1d - the effort to show the similarities between bulk- and single-cell expression data via correlation of F values is appreciated. However, there still seems to be a good portion of the correlation relationships that are either non-correlated or anti-correlated. This is not the most convincing figure to show the similarity between bulk and single-cell data. Is there some permutation analysis or statistical analysis to show that the number of F-correlations is indeed statistically

significant? If so, at what correlation level are the F-correlations between bulk and single-cell data statistically significant? It would be interesting to see what TF-TF relationships are anti-correlated or have no relationship and if that is indicative of some technical or biological factor.

We have calculated p -values for the correlation coefficient distributions using a Mann-Whitney U test to test if the medians deviate significantly from zero, and they are significant for all 3 subtypes. We have also inspected the correlations that are near-zero or negative in the revised manuscript and discussed their potential implications in our updated Discussion section.

- The patient stratification of low and high MYBL2 looks good in the Kaplan-Meier plots (Fig 5G), but the significance seems somewhat low (p -value (TCGA): 0.035 and p -value (CGGA): 0.0228).

The lower than anticipated significance is likely due to the relatively small sample sizes of these cohorts due to subsetting to the Proneural subtype. We are however confident that consistent observations in two completely independent datasets of two profiling platforms (micro-array and RNA-seq) and different demographics (largely non-Asian American and Chinese) are sufficient to prioritize MYBL2 for further investigation. Moreover, the p -value of <0.05 is a well-established cutoff for statistical significance.

- Ultimately, to be convinced, I would need some targeted experiments to validate new subtype-specific essential TFs.

We agree that *in vivo* validation experiments would be meaningful. However, we have been unable to perform such experiments due to COVID-related lab shut downs. We do, however, believe that the DepMap validation is compelling. It represents a lot of experimental manipulation of cell lines – work that is similar to what is requested here.

Minor comments:

- There are quite a few separate algorithms used as part of the overall workflow to generate a regulatory network model and predictions. It would be helpful if some type of flow-diagram were included, one that is a bit more detailed than the one included in Fig 1a. A supplementary figure that were to include in some way, shape, or form a general overview of the various algorithms, databases, and steps involved (e.g., PIQ algorithm, RANSAC algorithm, BIOGRID database, etc., etc., etc.).

- Supplementary Fig 1F is missing - no histograms showing overlap between TF-edges derived from TCGA and CGCA data are shown.

- Supplementary Fig 4C, which shows the percentile ranking of NFE2 essentiality across CL, PN, and MES subtypes, is not convincing. I would want to see a targeted essentiality test on NFE2 knockdown in different GBM subtype cells.

- Figure 5B axes are not labeled

- Supplementary Figure 1e legend - is "Mean symmetric mean absolute percentage error" correct? "Mean" is used twice in the sentence and may be a typo.

We appreciate these comments. All of these changes have been made and information provided in the revised manuscript, with exception of the experimental knockdown of NFE2.

Reviewer #3:

In this revised manuscript, the authors have adequately addressed several points raised in our earlier review. However, some key concerns remain unaddressed, which will be discussed in the following. First, let's start by acknowledging the points that have been addressed:

First, the authors now adequately explain why it was not viable to include cross-terms in their model. Also, while the use of RANSAC was questioned in the original review, the rebuttal clarifies how it was applied and shows that it is appropriate for their data. Finally, even though this was not specifically requested, in the original review, the new Figure S2e does a great job of illustrating how the in silico perturbational assay can move cells in gene expression space.

We very much appreciate the reviewer's comments.

This reviewer had also questioned the lack of comparative analysis of inTRINSiC to other established methods, such as ARACNe and SCENIC. As a result, it was difficult to weigh both benefits and drawbacks associated with the new algorithm and, in particular, to accurately assess whether it provided any improvements relative to the current state of the art. To address these concerns, the authors now utilize the BIOGRID database to explicitly compare the performance of ARACNe and inTRINSiC, concluding that they have comparable performances. They also clarify what they consider to be the specific advantages of their method. In particular, they illustrate inTRINSiC's ability to model the effect of downstream effectors on gene expression.

While these represents key improvements on the original manuscript submission, some key concerns remain unaddressed. For instance, the comparison between ARACNe and inTRINSiC relies largely on TF-TF interactions as a gold standard baseline, thus ignoring the vast majority of interactions which are between TFs and their transcriptional targets. BioGRID provides additional data for interactions between TFs and known targets, which could have been used for this purpose. There are also additional databases, such as TRRUST, which report information about TF-target interactions. Given the general availability of these data, why was the comparison performed using only a relatively restricted set?

We chose TF-TF interaction as a standard for evaluating model performance as it contains a large body of experimental evidence. In contrast, the BIOGRID data set is derived largely from predicted TF binding sites and physical binding experiments. Thus, our analysis focuses on a more extensively supported subset of the overall BIOGRID data set. We now describe this rationale in the revised manuscript.

While the authors discuss what they believe to be the advantages of the inTRINSiC approach relative to SCENIC in the discussion section, it appears that no direct comparison of their relative performance was made. As a result, for two of the mainstream approaches, it appears that one is essentially equivalent to the new methodology while the other was not tested at all.

A comparison with the general SCENIC approach is provided in the manuscript. When comparing our models with other methods, we have taken into consideration the major categories of regulatory network construction methods. SCENIC essentially runs a co-expression-based method (GENIE3) to unveil regulatory structures from single-cell expression profiles. We have also provided in our extended content (Figure EV1d and Materials and Methods) a comparison between our method and a linear regression-based method (Figure EV1c, EV2d).

More importantly, the authors have not explored any further validation of their downstream predictions, either experimentally or against other methods that identify key TFs that control cell state. This becomes more critical given that the ability of inTRINSiC to outperform other methods in terms of network reconstruction is not obvious from the new data. For instance, an obvious comparison would be to the SCENIC and VIPER algorithms which could also be benchmarked against DepMap, as done in Figure 5. This would establish an objective baseline for differential performance between inTRINSiC predictions and predictions generated by other methods. This is particularly critical because some of the predictions by inTRINSiC are not consistent with experimental validation. For instance, the algorithm predicts that silencing STAT3 in mesenchymal glioma samples would shift the phenotype towards a proneural phenotype. However, in the manuscript they cite (Carro et al. Nature 2010), experimental silencing of STAT3 alone failed to reprogram mesenchymal cells towards a proneural state and, as computationally predicted, joint silencing of both STAT3 and CEBPB was required to accomplish this goal.

We would like to emphasize that, as stated in our Discussion, other network inference algorithms, including SCENIC, does not allow for predictive gene expression modeling, i.e. given a perturbation, predict the changes in the expression levels of downstream regulatory targets. A comparison with the VIPER algorithm is indeed useful for evaluating the performance of the protein network activity prediction method in our paper. This is provided in the main text with results shown in Figure EV2h-j. When comparing protein signaling activity estimated from VIPER and that from our exponential ranking method, we found that exponential ranking was able to estimate protein activity changes for all 3,151 proteins covered by our signed, weighted model whereas VIPER only covered 218 proteins deemed as master regulators by the algorithm (Figure EV2h). Additionally, among the covered interactions, VIPER did not capture some of the well-established signaling activities, such as the EGFR/STAT3 and EGFR/KRAS axes, where EGFR is expected to activate these two downstream targets (Figure EV2i-j).

We believe the that reviewer may have misread our data here (or we have not explained it clearly). For the biology associated with STAT3, we would like to clarify that in our computational experiments, perturbing STAT3 caused a substantial shift towards the Classical subtype (Figure 3e) rather than the Proneural subtype (as stated by the reviewer). We also observed that some Proneural subtypes have also shifted towards the Mesenchymal subtype in Figure 3e upon loss of STAT3. We argue that this is consistent with two observations: 1) STAT3 has significant participation in both Mesenchymal and Proneural signature sets (Figure 3b), and 2) STAT3 is a part of the Mesenchymal master regulatory circuit and is required for maintaining the Mesenchymal state, which is shown in Carro et al (Nature 2010), Figure 4 where knocking down of either STAT3 or CEBPB would lead to a loss of Mesenchymal signature gene expression.

Similarly, while the example of MYBL2 is discussed at length, the authors do not provide any experimental validation to support these findings. Indeed, all validation attempts provided by the manuscript are completely retrospective, e.g. using DepMap data. As a result, contrary to what has been done by numerous manuscripts that have used computational approaches to prioritize cell dependencies or reprogramming factors and then validated these findings experimentally, this manuscript fails to provide any level of experimental validation supporting the claim that inTRINSiC predictions can discovering new biology.

As stated above, we agree that *in vivo* validation experiments would be meaningful. However, we have been unable to perform such experiments due to COVID-related lab shut downs. We do, however, believe that the DepMap validation is compelling. It represents and extensive experimental manipulation of cell lines – work that is similar to what is requested here.

Finally, I still feel that the authors do not adequately describe the real implications and value of inTRINSiC. The ability to perturb gene expression profiles in silico is novel and potentially interesting, but the authors present the algorithm as a way to prioritize essential TFs governing biological phenotypes and that would require either experimental validation to substantiate their claims or comparison to experimentally validated algorithms that perform similar tasks. As a result, proposing that inTRINSiC may be able to predict treatments entirely in silico seems tenuous and unsubstantiated by the data, especially given the difficulty of directly targeting TFs, a point the authors concede in the rebuttal, but do not discuss at all in the revised manuscript.

We have perhaps understated the relevance and utility of inTRINSiC in the manuscript. The framework for predictive modeling of gene essentiality provided by inTRINSiC will not only serve as a powerful platform for predicting treatment outcome in silico, but will also prove highly valuable in at least two areas: First, it will allow one to leverage large-scale profiling datasets from primary human tumor samples (GBM and otherwise) where unbiased genetic screens are infeasible to perform. This analysis will identify “high value” genes and pathways to target in cell line or mouse/xenograft models. We are unaware of other approaches that can effectively do this. Second, with improvements in the scale and coverage of newer techniques such as Perturb-seq and single-cell epigenomics, the inTRINSiC pipeline can be extended to understand the source of phenotypic heterogeneity at the single-cell level through modeling multiple layers of regulation in the same cell.

Finally, while targeting TFs themselves is challenging, such efforts are increasingly common and, as in the case of Brd4 inhibitors, represent a clinical reality.

All of these points are now included in the revised manuscript.

Manuscript Number: MSB-20-9506RRR, Integrated regulatory models for inference of subtype-specific susceptibilities in glioblastoma

Thank you for sending us your revised manuscript. We think that the performed changes and the explanations provided in the point by point response satisfactorily address the remaining issues raised by the reviewers. As such, I am glad to inform you that we can soon accept your manuscript for publication, pending some minor editorial issues listed below.

The authors performed the requested changes.

Manuscript number: MSB-20-9506RRRR, Integrated regulatory models for inference of subtype-specific susceptibilities in glioblastoma

Thank you again for sending us your revised manuscript and for performing the last requested edits. We are now satisfied with the modifications made and I am pleased to inform you that your paper has been accepted for publication.

Corresponding Author Name: Michael T. Hemann

Manuscript Number: MSB-20-9506R